



# 1 A Canadian River Ice Database from National Hydrometric Program
# 2 Archives

Laurent de Rham[1], Yonas Dibike[1], Spyros Beltaos[2], Daniel Peters[1], Barrie Bonsal[3], Terry Prowse[1]
[1]Environment and Climate Change Canada, Watershed Hydrology and Ecology Research Division, 3800 Finnerty Rd.,
Victoria, BC V8P 5C2, Canada
[2]Environment and Climate Change Canada, Watershed Hydrology and Ecology Research Division, 867 Lakeshore Rd.,
Burlington, ON, L7S 1A1, Canada
[3]Environment and Climate Change Canada, Watershed Hydrology and Ecology Research Division, 11 Innovation Blvd.,
Saskatoon, SK, S7N 3H5, Canada
*Correspondence to*: Laurent de Rham (laurent.derham@canada.ca)

**15 Abstract**

River ice is a common occurrence in cold climate hydrological systems. The annual cycle of river ice formation, growth, decay
and clearance can include low flows and ice jams, as well as mid-winter and spring break-up events. Reports and associated
data on river ice occurrence are often limited to site and season-specific studies. Within Canada, the National Hydrometric
Program (NHP) operates a network of gauging stations with water level as the primary measured variable to derive discharge.
In the late 1990s, the Water Science and Technology Directorate of Environment and Climate Change Canada initiated a long-
term effort to compile, archive and extract river ice related information from NHP hydrometric records. This data article
describes the original research data set produced by this near 20-year effort: the Canadian River Ice Database (CRID). The
CRID holds almost 73,000 variables from a network of 196 NHP stations throughout Canada that were in operation within the
period 1894 to 2015. Over 100,000 paper and digital files were reviewed representing 10,378 station-years of active operation.
The task of compiling this database involved manual extraction and input of more than 460,000 data entries on water level,
discharge, date, time and data quality rating. Guidelines on the data extraction, rating procedure and challenges are provided.
At each location, a time series of up to 15 variables specific to the occurrence of freeze-up and winter-low events, mid-winter
break-up, ice thickness, spring break-up and maximum open-water level were compiled. This database follows up on several
earlier efforts to compile information on river ice, which are summarized herein, and expands the scope and detail for use in
Canadian river ice research and applications. Following the Government of Canada Open Data initiative, this original river
ice data set is available at: https://doi.org/10.18164/c21e1852-ba8e-44af-bc13-48eeedfcf2f4 (de Rham et al., 2020)

**34 1 Introduction**





River ice and ice-related events are a common feature throughout cold-climate regions.  However, the hydrological and hydraulic effects of ice receive considerably less attention than open-water river conditions.  In the past decade, the study of river-ice processes and hydraulics emerged as an important research area (Hicks, 2008) with a renewed focus on ecological aspects (e.g. Peters et al., 2016; Lindenschmidt et al., 2018). Given recent rapid changes to the cryosphere, there is a need to better understand river ice processes and hydraulics as they relate to a warming climate (Derksen et al., 2019). Advances in river ice process science are largely driven by observation and collection of field data supplemented by hydraulic modelling. However, most studies have been limited to a specific location or river reach and focused on a particular part of the ice period, such as the spring break-up. It is not well known by the hydrologic research community that a valuable source on river ice information can be extracted from the archives of hydrometric networks. In Canada, the National Hydrometric Program (NHP), in partnership with the Water Survey of Canada (WSC), provinces and territories, operates a current network of more than 2,800 hydrometric stations covering a broad range of hydroclimatic and hydrologic conditions, thus providing a good cross-section of the various river ice types and regimes.  Historically, the primary mandate of the NHP was to provide water quantity information published as a time series of river discharge. The associated water level data, a requisite for calculation of discharge, has not been published up until the turn of this century. Importantly, the NHP accounts for the hydraulic effects of ice on river channels when calculating discharge. Archival data used to compute discharge values in the form of field site visit notes, occasional ice thickness measurements, and continuous water level records, are a valuable source of information for the scientific, engineering and water management communities.

The Committee on River Ice Processes and the Environment (CRIPE; http://www.cripe.ca/) sponsored report *Working Group on River Ice Jams - - Field Studies and Research Needs* by Beltaos et al., (1990) includes a chapter with detailed guidelines on the extraction of river ice data from hydrometric archives.  Although field observations and data can be imperfect, with evidence of ice recorded only to improve the hydrometric program's discharge estimates, the archives cover a range of locations and are accessible upon request. Based on these beneficial attributes, efforts towards the creation of a database of river ice parameters were recommended (Beltaos, 1990) and a compilation of the hydrometric archives for a pan-Canadian river ice database began in the late 1990s. Prowse and Lacroix (2001) reported on the extraction of spring break-up extreme events at a subset of 143 NHP gauging sites up to the year 1999, covering major drainage basins and ecological zones in Canada. This work was followed by a preliminary analysis on 111 sites proximal and north of the annual 0°C isotherm, differentiating between ice-induced and open-water flood generating mechanisms (Prowse et al., 2001). von de Wall et al., (2009, 2010) also used  NHP sites north of the temperate ice zone, covering the years 1913 to 2006, for analysis of the spring break-up period. These works reported on the geographical distribution and statistical analysis of physical controls on flood generating mechanisms, a trend analysis (1969-2006), as well as correlations of ice event occurrence to both the 0°C isotherm and various atmospheric teleconnection patterns.





More common in Canada are watershed and reach-scale studies of river ice processes. Examples include the work of de Rham
et al., (2008a, 2008b) who examined spatial and temporal characteristics of the timing and magnitude of the spring break-up
period from 1913 to 2002 throughout the Mackenzie River Basin. Downstream in the Mackenzie River Delta, Goulding et al.,
(2009a, 2009b) assessed spring break-up and ice jam water level event timing and magnitude to provide insights on hydro
climatic controls of the break-up sequence over the 1974-2006 period. For the upstream Peace watershed, Beltaos (2003a,
2003b) and Beltaos and Carter (2009) utilized field based data and hydraulic modelling to examine the effects of hydroelectric
reservoir operation on fall freeze-up and spring break-up flows and levels in the lower Peace River; the objective was to address
the question of declining ice-jam flooding of the Peace-Athabasca Delta (Beltaos, 2018), while Peters et al., (2006) examined
the maximum extent of flooding of ice-jam vs open-water flood events in this delta.

Expanding beyond Canada, Newton et al., (2017) reported on hydro-climatic drivers on mid-winter break-up occurrence
derived from NHP hydrometric records for western Canada and the Cold Regions Research and Engineering Laboratory Ice
Jam Database (IJDB) for Alaska (1950-2014). The IJDB (Carr et al., 2015) includes the timing and magnitude of ice-jam
events across the United States for the period 1780 to present. While data sources are wide in scope, the initial creation of the
IJDB during the 1990s drew largely from the United States Geological Survey (USGS) gauging station data, including peak
backwater level events (White, 1996). Outside of North America, efforts to compile river ice information from hydrometric
data have included work to assess river break-up dates (1893-1991) in Russia (Soldatova, 1993). The National Snow and Ice
Data Centre (NSIDC) provides online access to Russian River Ice Thickness and Duration (1917-1992) dataset (Vuglinsky,
2000). These databases have been used for assessments of river ice conditions (e.g. Smith, 2000; Vuglinsky, 2006), with select
at-site updates to the year 2012 (Shiklomanov and Lammers, 2014). The NSIDC also provides access to The Global Lake and
River Ice Phenology Database, Version 1 (Benson et al., 2000) that includes time series of freeze, thaw/break-up dates and
description of ice cover for 237 rivers. Although not specific to river ice processes, the national scale Canadian Ice Database
(CID; Lenormand et al., 2002) also compiled visual observations of freeze-up and break-up dates along with measurements of
ice thickness at 288 rivers across Canada. Brooks et al., (2013) used the data from the CID, along with international and NHP
archives to quantify freshwater ice characteristics in the Northern Hemisphere.

Beltaos and Prowse (2009) presented a comprehensive review of global changes in river ice processes. While overall results
indicated a shortening ice season, the authors noted that the majority of published studies assessed freeze-up and break-up
dates, which can be more readily obtained from hydrometric agencies, rather than the more difficult to obtain daily and
instantaneous ice-affected water levels. Specifically, broad-scale studies assessing river ice data extracted directly from
hydrometric archives are yet to be completed. Thus, only a limited body of published research is available assessing the
magnitude and timing of specific, dynamic river ice variables during the fall freeze-up, mid-winter, winter-low and spring
break-up periods.



This paper expands upon the brief overview of the Canadian River Ice Database (CRID) presented at CRIPE (de Rham et al.,
2019) and aims to provide a comprehensive reference document to accompany the publication of the CRID on the Government
of Canada Open Data Portal. The main objectives are to:  1) describe the NHP archives and data collection history of this
study; 2) present the 15 variables identified from the NHP archives recordings outlining the data extraction procedure; 3) report
on challenges, assumptions and uncertainties encountered in the extraction of river ice information from hydrometric archives;
and 4) identify resource requirements if others elect to undertake similar effort and highlight potential uses for this river ice
database. The paper begins by describing the Study Area and Hydrometric Monitoring Sites followed by the Methodology
covering details of the data extraction procedure.  The Discussion section summarizes the data and highlights database utility
and future research needs. The paper ends with sections on Data Availability, Data Disclaimer and Conclusion.

**2 Study Area and Hydrometric Monitoring Sites**

The locations and characteristics of NHP stations, including their operation and regulation history, are available (in
downloadable .csv format) at: https://wateroffice.ec.gc.ca/station_metadata/reference_index_e.html.  The CRID includes data
on river ice affected water level, associated channel flows and timing at a subset of 196 gauging stations across Canada (Fig.
1). The monitoring sites are located within 11 of the 13 provinces and territories, and extend over 10 of the 11 Canadian climate
regions (Gullet et al., 1992). In the beginning, the database focused on 143 stations with a minimum 20-year record, drainage
area greater than 10,000 km$^2$, and located north of the mean annual 0°C isotherm (Prowse and Lacroix, 2001).  Thereafter, an
examination of spring break-up at 136 northern gauging sites was reported (von de Wall, 2011). For the current study, the
geographic criterion was expanded south into a "temperate zone" (Newton et al., 2017) and the drainage area threshold was
removed. A review of literature and correspondence with WSC staff and provincial flood authorities identified an additional
60 southern sites prone to mid-winter break-up events. The database now includes 196 sites with drainage areas ranging from
20.4 km$^2$ to 1.68 x 10$^6$ km$^2$, includes both natural and regulated flow conditions, with the latter distributed throughout this
range.

The flow regime at the 150 natural sites has not been affected by any significant upstream waterworks. At the remaining 46
regulated gauging stations, predominantly in southern Canada (Fig. 1), flows were affected by instream waterworks, such as
weirs, dams and water diversion/abstraction. The majority of natural sites (120) were in operation up to the end of the study
period of Dec 31, 2015, while most of the discontinued (30) stations ceased operating in the mid 1990s (Fig. 2). This late 20$^{th}$
century reduction in the monitoring network has also been reported by others (Lenormand et al., 2002; Lacroix et al., 2005).
The regulated sites include 29 homogeneous (entire period of operation regulated) and 17 heterogeneous (natural then regulated
flow during period of operation) hydraulic conditions (Fig. 2).  The Peace River system, an example of a heterogeneous
hydrometric archive, is affected by both climate and regulation and a system of hydro-ecological foci (e.g. Hall et al., 2018;
Timoney et al., 2018; Beltaos, 2019).  A large number of the older stations have periods of inactive operation during 1920 to
1960. A few inactive stations resumed operation since shutdown in the mid-1990s (Fig. 2). After removing the 1,012 years of inactive status, the 196 NHP sites considered represent 10,378 station-years of data prior to 2016. Appendix A1 provides a listing of all the stations selected for the CRID, including start and end dates and type. Specific CRID locations within this paper are referenced by gauging site name followed by the NHP alpha-numeric identifier in brackets.

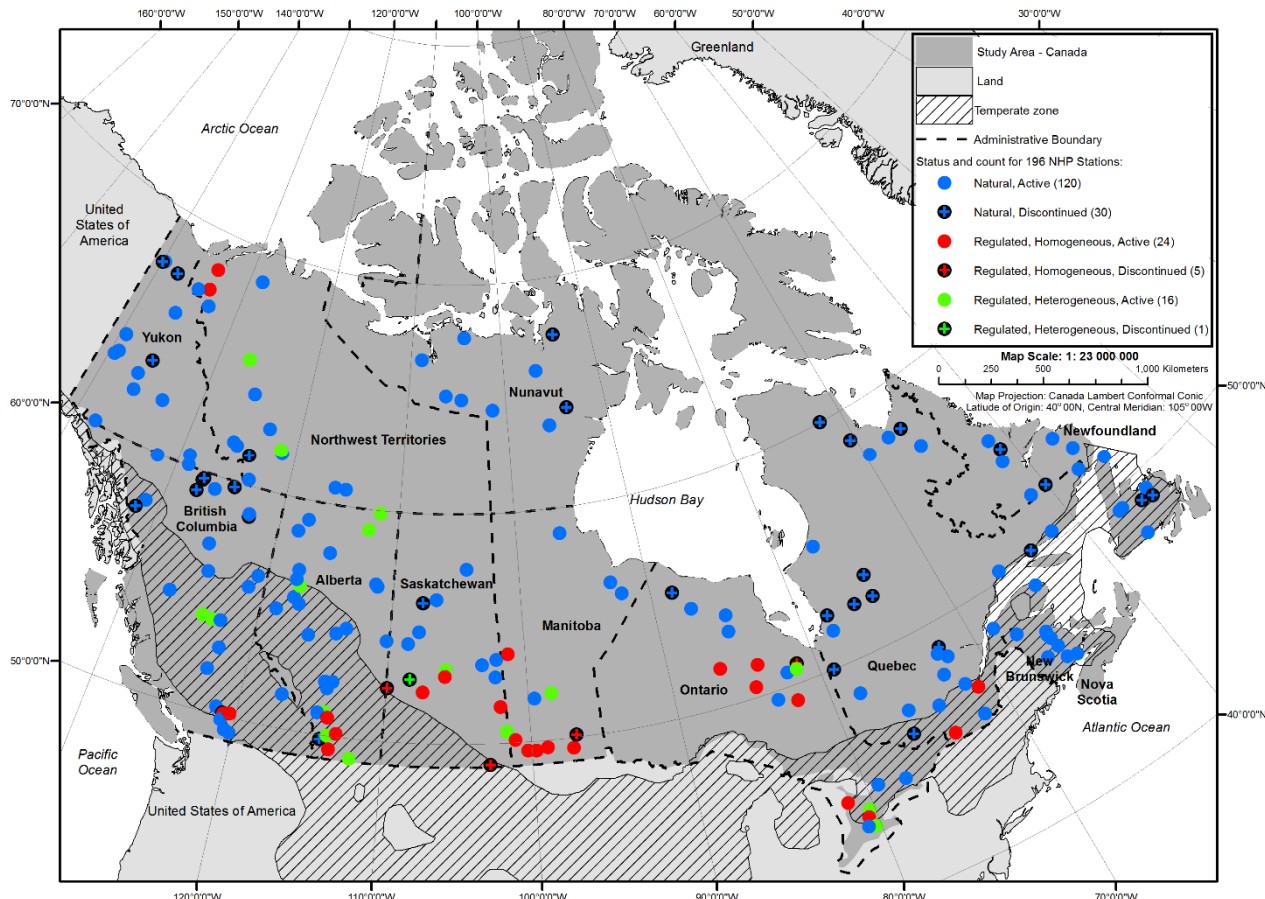

**Figure 1.** Location of the 196 National Hydrometric Program (NHP) hydrometric gauging stations included in the Canadian River Ice Database. Status and count for the stations are based on flow condition (Natural or Regulated), Active (in operation up to end of 2015) or Discontinued and if flow condition is homogeneous (always regulated) or heterogeneous (regulated during specific period of operation).

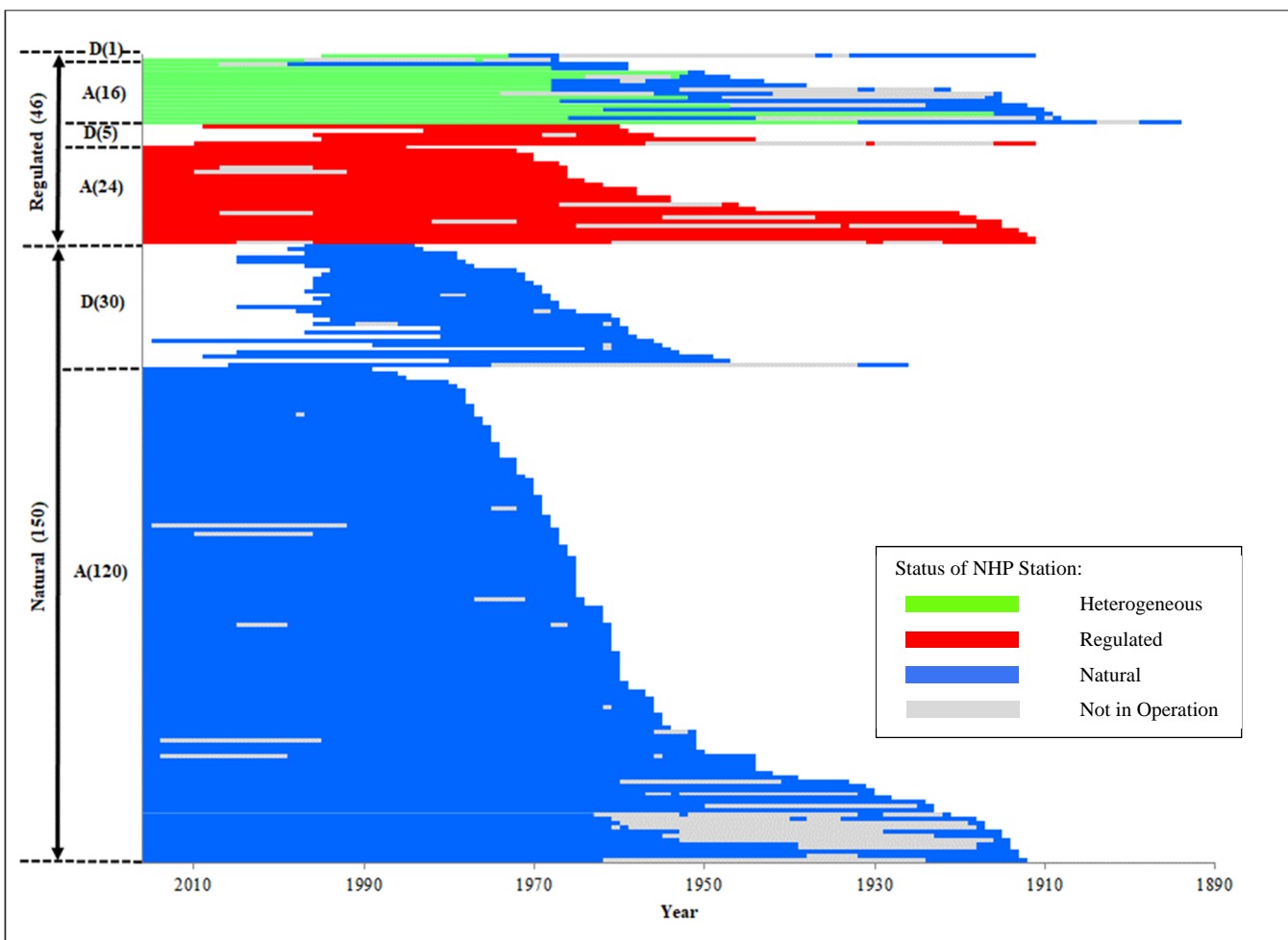

**Figure 2.** Bar chart showing the operational history of the 196 National Hydrometric Program (NHP) included in the Canadian River Ice Database. Stations are categorized by flow conditions (Natural or Regulated), homogeneity in flow conditions (homogeneous or heterogeneous), and operational status (Active (A) or Discontinued (D)). The number in each sub-category is shown brackets.

# 3 Methodology

## 3.1 National Hydrometric Program Archives

The specific paper and digital hydrometric archives compiled and reviewed for this study include: (1) continuous water-level pen recorder charts (before year ca. 2000) during the freeze-up, mid-winter break-up (if applicable) and spring break-up periods; (2) digital files (after year ca. 2000 onwards) with water level data at discrete 5- to 15- to 60-minute interval, some



including minimum and maximum instantaneous water level for entire annual period; (3) station descriptions; (4) site visit
survey notes, including ice thickness summary files; (5) gauge and benchmark history; (6) stage-discharge (S-Q) relationship
tables; (7) annual station analyses; (8) annual water level tables; (9) discharge measurement summaries; and (10) yearly
station summary files (year ca. 2003-2009). Archives since 2009 are in digital format extracted from the Aquarius water data
management platform, which simplified the data extraction, as compared to reading hand-written notes and pen charts for prior
years. The end year 2015 was selected for the CRID as finalized NHP archival data can be delayed by up to two years while
data control protocol is followed. The NHP works with provincial governments and partner organizations at some network
stations - archives also include those provided by the governments of Alberta, Saskatchewan, as well as the Centre d'Expertise
Hydrique du Quebec (CEHQ). An earlier report (Groudin, 2001) included baseline break-up and open-water river information
for 16 Quebec sites. Supplementary digital daily water level data for Quebec stations (Table A1; stations with "RIVIERE" in
name) prior to ~ 1997 were limited to first water level recording of the day and, thereafter, summaries of 15 minute and daily
average water level were provided. Information on discharge and river ice data qualifiers (such as the B dates, discussed below)
were gleaned from the following WSC and CEHQ internet sites: https://wateroffice.ec.gc.ca/index_e.html and
http://www.cehq.gouv.qc.ca/hydrometrie/index-en.htm.

The evolution of the CRID was comprised of six data collection campaigns since 2000 (Table 1). Major data archival efforts
in the years 2000-01 and 2010-11 required a team of two to three people visiting up to 8 WSC regional offices, with each visit
lasting up to 2 weeks to photocopy and/or scan hydrometric archives. Following that, all paper based information, except for
Quebec stations, was digitally scanned and filed to a central electronic repository. This 0.5 Terabyte digital data entity consists
of over 30,000 folders and 100,000 files that is currently stored on a secure Environment and Climate Change Canada server.
The CRID digital archive is available on request.

**Table 1.** List of the six data collection campaigns towards the development of the Canadian River Ice Database. The Water
Survey of Canada (WSC) is the federal part of the National Hydrometric Program (NHP), which also includes provincial and
territorial agencies.



| Data Collection Campaign | Study Focus | Location of NHP Sites | Number of NHP sites | NHP site Archival & Extraction | WSC Regional Office Visits | Duration of Office Visit | WSC Regional Office Locations and NHP partners | Publications |
|---|---|---|---|---|---|---|---|---|
| 2000-01 | spring break-up | Northern Canada | 143 | up to 2001 | 8 | up to 2 weeks | Vancouver, Calgary, Yellowkinfe, Regina, Winnipeg, Burlington, St. Johns, Cornerbrook; Groudin (2001) report on Quebec sites | Prowse and Lacroix 2001; Prowse et al., 2001 |
| 2003 | spring break-up | Mackenzie River Basin | 29 | 2002 | 5 | up to 1 week | Inuvik, Fort Simpson, Calgary , Inuvik, Peace River | de Rham 2006; de Rham et al., 2008a, 2008b |
| 2007 | spring break-up | Mackenzie River Delta | 14 | 2002-2006 | 2 | up to 1 week | Yellowknife, Inuvik | Goulding 2008; Goudling et al., 2009a, 2009b |
| 2008-2009 | spring break-up | Northern Canada | 136 | 2002-2006 | - | - | transfer of digital information from 8 regional offices and 3 provincial agencies | von de Wall et al., 2009, 2010; von de Wall 2011, |
| 2010-2011 | fall freeze-up, mid-winter and spring break-up | Canada | 196 | up to 2008 | 7 | up to 2 weeks | Vancouver, Calgary, Yellowkinfe, Regina, Winnipeg, Burlington, Fredericton; digital information from 3 provincial agencies | Brooks, 2012; Brooks et al., 2013, Newton et al, 2017; Newton, 2018 |
| 2017-2018 | fall freeze-up, mid-winter and spring break-up | Canada | 196 | 2009 - 2015 | - | - | transfer of digital information from 7 regional offices and 3 provincial agencies | de Rham et al., 2018, de Rham et al., 2019 |

## 3.2 Data Extraction and Quality Rating

A conceptual schematic of a water level hydrograph showing all typical ice effected metrics is plotted in Fig. 3. The CRID includes up to 15 variables extracted from NHP recorded archives that cover the water year (Table 2). These variables are categorized as occurring during the: freeze-up, ice cover, break-up, or open-water season. For the variables shaded in grey, the objective was to record data on instantaneous water level, associated date and time. These instantaneous values reflect the maximum flood potential. The procedure for extracting river ice data follows the guidelines of Beltaos (1990), and primarily involves visual examination of water level records. Hence, identification and extraction of river ice data is a subjective process and the accuracy to which water level, discharge and event timings were registered is included in Table 2. Depending on the possibility of extracting instantaneous (Table 2, grey shading), daily water level or discharge ($H_{LQ1}$, $H_{LQ2}$) based variable, a data quality rating scheme with values of 0, 1 and 2 was used to quantify the continuum of higher to lower data accuracy (Table 3). Under some circumstances, judgement was applied to rate data quality higher or lower depends on various circumstances, such as termination of a continuous water level record during the spring break-up season where ice movement, synonymous with variable spring break-up initiation (Sect. 3.4.6) damaged the recording instrument. Such data would rate as 0 even though data from the fragmented record rates as 1 on Table 3.


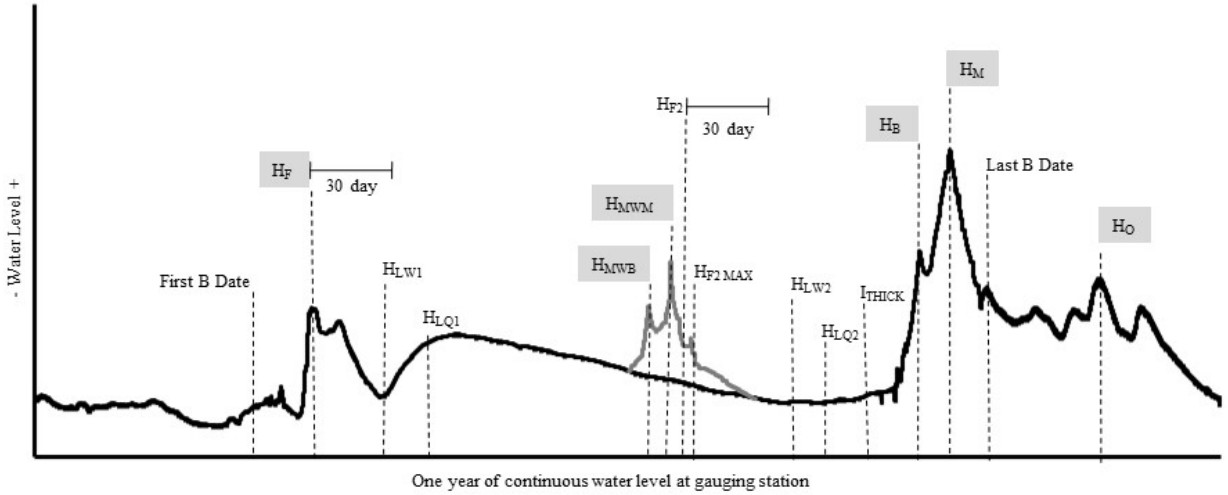


**Figure 3.** Conceptual schematic of continuous river water level hydrograph (black line). Possible mid-winter break up event shown as grey line, at approximate center of hydrograph. Symbols for the 15 variables which populate the Canadian River Ice Database are shown in the figure (see Table 2 for additional information). For the variables shaded in grey, the objective was to record the instantaneous water level and associated time when the event occurred.

**Table 2.** The 15 variables extracted from the National Hydrometric Program archives and input to the Canadian River Ice Database (CRID). The CRID includes the date of all variables classified by season. The accuracy to which the water level or discharge record was examined is summarized with grey shading denoting attempt to identify instantaneous water level events. Data quality rating was applied to the underlined data.



| Season | Variable | Symbol | Description | Data Accuracy: Instantaneous (I), Daily (D), No Extraction (-) | | | Data Quality Rating (0-1-2) |
| | | | | Water Level | Discharge | Time | Yes (Y) or No (N) |
| --- | --- | --- | --- | --- | --- | --- | --- |
| Freeze-up | First Day With Backwater Due To Ice | First B Date | First day that ice affects channel flow conditions | - | - | D | N |
| Freeze-up | First Freeze-Over Water Level | $H_F$ | Channel wide ice cover; daily water level at $H_F$ and following 29 days | I or D | D | I or D | Y |
| Ice cover | First Minimum Winter Water Level | $H_{LW1}$ | Minimum daily water level between $H_F$ and $H_B$ | D | D | D | Y |
| Ice cover | First Minimum Winter Discharge | $H_{LQ1}$ | Minimum daily discharge between $H_F$ and $H_B$ | D | D | D | Y |
| Ice cover | Mid-Winter Break-Up Initiation | $H_{MWB}$ | Initiation of mid-winter break-up event | I or D | D | I or D | Y |
| Ice cover | Maximum Mid-Winter Break-Up Water Level | $H_{MWM}$ | Maximum mid-winter break-up event water level | I or D | D | I or D | Y |
| Ice cover | Maximum Winter Water Level | $H_{F2}$ | Freeze-up after $H_{MWM}$. If no Mid-winter event, first day of 7 day average if exceeds $H_F$ 7 day average | D | D | D | Y |
| Ice cover | Maximum Winter Water Level 7 Day | $H_{F2\,MAX}$ | Maximum daily water level within first 7 days following $H_{F2}$ | D | D | D | Y |
| Ice cover | Second Minumum Winter Water Level | $H_{LW2}$ | Minimum daily water level between $H_{F2}$ and $H_B$ if $H_{LW1}$ before $H_{F2}$ | D | D | D | Y |
| Ice cover | Second Minimum Winter Discharge | $H_{LQ2}$ | Minimum daily discharge between $H_{F2}$ and $H_B$ if $H_{LQ1}$ before $H_{F2}$ | D | D | D | Y |
| Ice cover | River Ice Thickness | $I_{THICK}$ | Average channel ice thickness prior to spring break up | - | - | D | N |
| Break-up | Spring Break-Up Initiation | $H_B$ | Begining of spring break up event | I or D | D | I or D | Y |
| Break-up | Maximum Spring Break-Up Water Level | $H_M$ | Maximum spring break-up water level event | I or D | D | I or D | Y |
| Break-up | Last Day With Backwater Due To Ice | Last B Date | Final day that ice affects channel flow conditions | - | - | D | N |
| Open-Water | Maximum Open-Water Level | $H_O$ | Maximum water level occuring outside First B date to Last B date | I or D | I or D | I or D | Y |

**Table 3.** The data quality rating for water level or discharge associated with 12 of the 15 variables in the Canadian River Ice Database. Continuous indicates no gap in the recorded hydrometric data, fragmented means there are some gaps over the period of review, and sporadic indicates limited data available. This was a qualitative, expert judgment-based rating.





| Data | Data Quality Rating | | |
|------|---|---|---|
| | 0 | 1 | 2 |
| Instantaneous Water Level | continuous | fragmented, continuous daily | fragmented daily |
| Daily Water Level or Discharge | continuous | fragmented | sporadic |

### 3.3. Ice Affected Stage-Discharge Relationship and B Dates

This section highlights challenges related to data collection during the ice season through excerpts from hydrometric program operational manuals, other publications and experience in developing this database. This background information is considered of high value to users when interpreting spatial and temporal characteristics of river ice.

A fundamental concept in hydrometry is the stage – discharge (S-Q) relationship.  At each NHP monitoring location, a reach-specific relationship is established via field surveys. Each year, hydrometric staff complete multiple site visits to measure in situ stream velocity and flow area to calculate discharge for a given water level. This work is ongoing with occasional refinement and adjustment of the S-Q relationship to account for changes in channel morphology and bed roughness – in some cases requiring relocations of station due to loss of stable control section in response to natural and/or anthropogenic impacts. Besides, the open water S-Q relationship is not valid during river ice conditions due to well-known hydraulic effects of ice on flow conveyance.  In Canada, ice-influenced flows are identified with a "B" flag to inform the user that the water level is affected by 'Backwater' conditions leading to a higher water level associated with a given discharge on the S-Q curve.  The specific river ice condition can take different forms, such as frazil and slush ice, anchor ice, partial ice cover, complete ice cover, ice jams, flowing ice chunks or a mix of these (Poyser et al., 1999). The data user, therefore, has to be aware of these possibilities when using 'B' dates as metric for river ice conditions. In reference to S-Q relationships under ice, Environment Canada (1980) states: *"Because of the many variable factors involved, no single standard procedure is suggested for the computation of daily discharges during periods when the stage-discharge relation is affected by the presence of ice. Several methods of computing discharges under ice conditions are available and it is suggested that the Regional Offices use the method that best suits each individual station"*. The CRID, with data sourcing from regional offices and partner organizations across the country, inherits this discharge calculation legacy for the 12 reported ice affected discharge time series (Table 2). Cold-region hydrometric programs have to contend with measurement problems and uncertainties of under-ice flows (Pelletier, 1990). Accurate measurement receives continued attention since water resource managers, dam operators and the flooding research community seek to reduce uncertainty (e.g. Healy and Hicks, 2004; Fulton et al., 2018) for ice affected periods. The apparently chaotic flow condition during the freeze-up and break-up periods along with Kennedy's (1975) observation that:



*"an ice-jammed river is among the most deranged of hydraulic phenomena"* further complicate discharge estimation.  The
WSC Lesson Package No. 20 – Computation of Daily Discharge (Ice Conditions) (Poyser et al., 1999) reiterated freeze-up and
break-up as: "*two periods are often the most difficult ones for which to produce reliable discharge estimates, even for seasoned*
*hydrometrists, who must use ingenuity, experience, and a knowledge of the characteristic traits that indicate transition*" and
that "*Computation under ice conditions involves a high level of personal judgement on the part of the technician in the*
*interpretation of the available data*".

Thus, interpretation of ice affected conditions remains a challenge for hydrometric programs. For example, at a gauge station
along the Peace River (https://wateroffice.ec.gc.ca/report/historical_e.html?stn=07KC001) the WSC informs users ""*Data*
*quality during spring break-up considered poor and remaining ice period considered fair*". Background for this assessment
is provided by Fig. 4, in which the latest time when ice-covered flow can be estimated with a fair degree of confidence is at
point A. Under conditions of a stable ice cover, hydrometric staff can apply site-specific methods to estimate the applicable
discharge, based in part on sporadic flow measurements during the winter period. Point B in Fig. 4 denotes the last day of
backwater, so that after that time discharge can be estimated with very good confidence using the gauge-specific S-Q
relationship that applies to open-water conditions. Point C in Fig. 4 approximately delineates the periods of pre-breakup
(sheet ice cover, possibly subjected to hinge and transverse cracking) and actual breakup when various events such as ice
jams and ice runs generate repeated increases and decreases in the water level that are too sharp to be runoff-generated. For
the breakup period, hydrometric staff estimate daily flows by taking into account the general trend of the water level
hydrograph, prevailing weather conditions, flows at upstream gauges and tributaries, as well as any in-situ visual
observations that may be available. Once the ice cover is fractured, mobilized, and broken up, flow measurement is inhibited
by problematic access and safety considerations. Consequently, it is not possible to assign error margins to associated flow
estimates, leading to the aforementioned "poor" characterization.






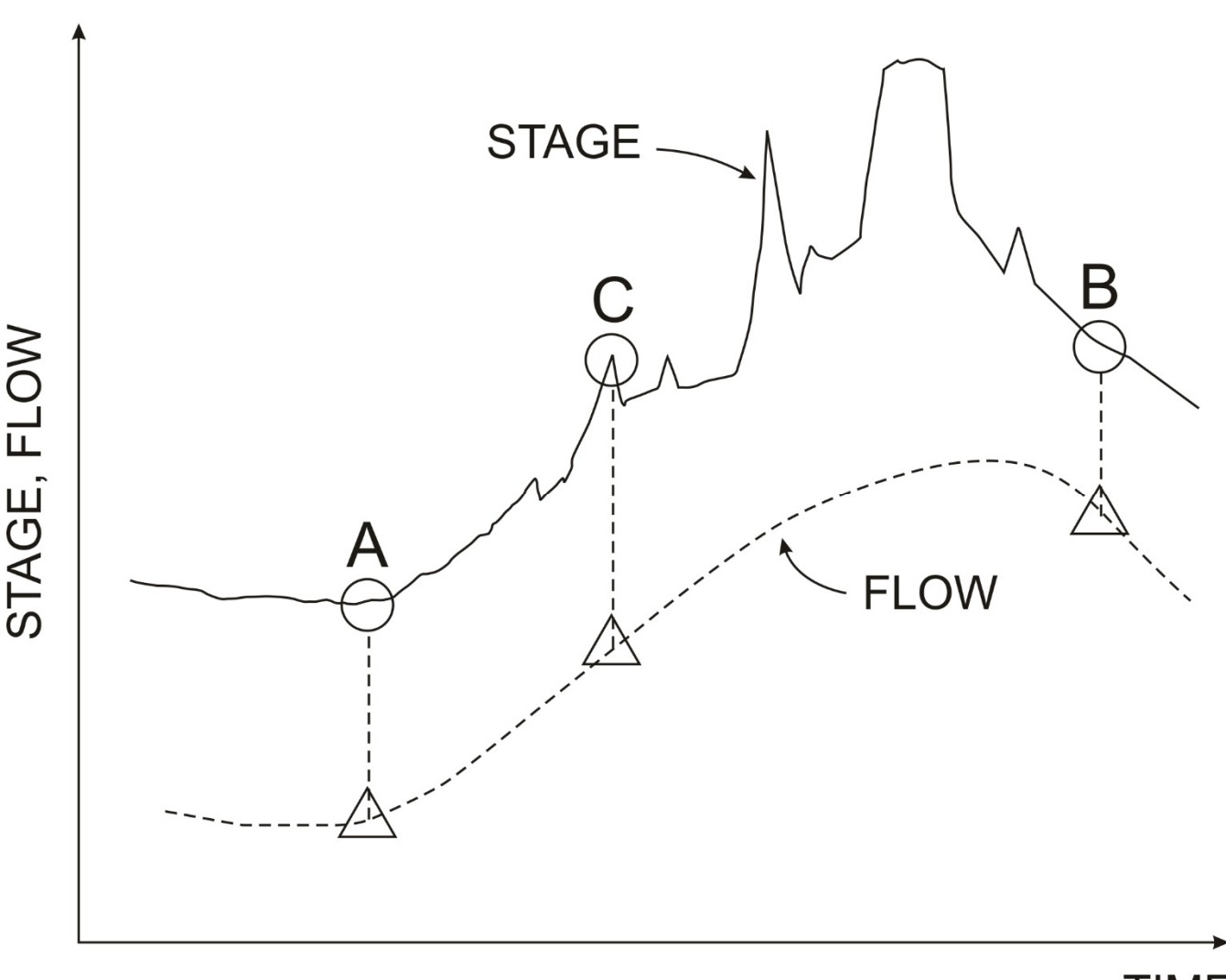


**Figure 4.** Schematic illustration of typical stage (i.e. water level) and flow (i.e. discharge) variations during the early phase

of the spring runoff event. From Beltaos (2012); Crown Copyright; Published by NRC Research Press.

The first ever published analysis of WSC 'B' dates was completed by Brimley and Freeman (1997) who examined trends in
the Atlantic region. Their observations on station locations and the dynamic ice conditions *"that the data on river ice should*
*only be considered valid at the gauging station site and may not be transferable to the entire watershed"* are applicable to
the CRID product.





Users of ice-affected discharge estimates are encouraged to actively report the data uncertainties inherent to the ice period
and how station location and hydraulic conditions can affect the ice and flow regimes.  This practice informs the water
community on a unique characteristic of cold-regions hydrometry and caution in interpreting study results. As a corollary,
the water level interpretation toward the CRID research data set also required a high level of expert judgement with this
subjective attribute inherent to the reported variables.

The following sub sections, corresponding to the season of occurrence (Table 2) aims to provide the background, extraction
details and literature justifications for the CRID variables.

**3.4.1 Freeze-up: First B Date, $H_F$**

As mentioned above, the NHP daily discharge values include a 'B' date flag to inform users of discharge estimates that consider
the ice "Backwater" effect in the stream reach (Environment Canada, 2012). Users can access these data in the online archive
and/or downloadable HYDAT database with the Environment Canada Data Explorer freeware
(https://wateroffice.ec.gc.ca/mainmenu/tools_and_downloads_index_e.html). The first occurrence of this flag, the First B
Date, marks the beginning of ice affected channel flow condition and has been used to investigate changes in the timing of
river freeze-up (Zhang et al., 2001; Peters et al., 2014). However, the First B Date does not indicate the presence of an ice
cover at a hydrometric gauge since the backwater effect may be a result of ice conditions far downstream of the station or
nearby presence of significant anchor ice build-up on the river bed. The MODIS time-lapse satellite images in Fig. 5 illustrate
the freeze-up and ice cover conditions on a reach of the Mackenzie River in the fall of 2000. For that year, NHP reports a First
B Date of Oct 10, but open water sections appear on Oct 14 and even one month later on Nov 7.  Only the Nov 12 image shows
the ice cover over the entire river channel with no open water sections apparent. The First B Date in the CRID therefore only
marks the beginning of ice effects on a river reach and cannot be assumed to be a channel wide ice cover condition. Though
extraction of CRID variables did not use alternative means of verification, using satellite images from the WorldView interface
(accessed at: https://worldview.earthdata.nasa.gov/) in this example is a simple way to view time series of changing ice cover
conditions since the year 2000.  For locations with several freeze-up and break-up cycles, such as the temperate zone locations
(Fig. 1) or gauges with associated intermittent daily B data flags (depicted on Fig. 9, Sect. 3.4.4 ), the first B occurrence was
recorded as First B Date.  For CEHQ stations in Quebec, the data qualifier R was assumed synonymous to B and in the very
few situations where the date did not match, NHP First B Date was used.

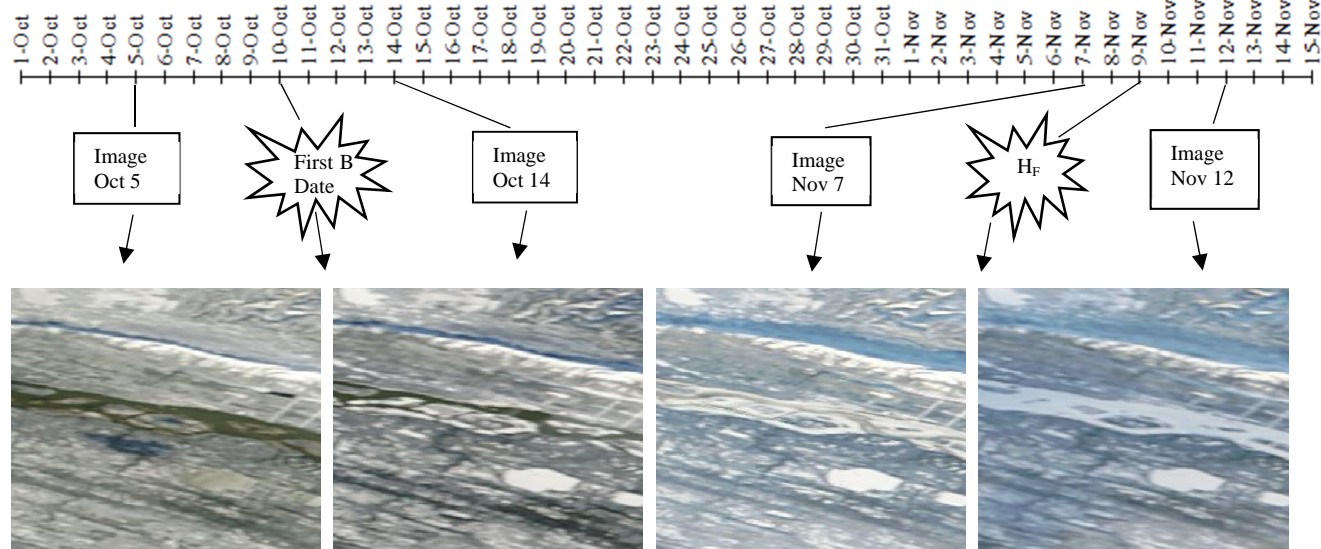



**Figure 5.** Year 2000 MODIS time-lapse satellite images (accessed at: https://worldview.earthdata.nasa.gov/) at the National Hydrometric Program gauge location Mackenzie River at Norman Wells (10KA001). Station is located near centre of the images. Width of the channel is approximately 1,300 meters and includes numerous islands. Flow is from right to left. First B Date is October 10 while freeze-over water level ($H_F$) occurred November 9 and open water appears during the freeze-up season. Images on First B date and $H_F$ were obscured by clouds.

Formation of a channel-wide ice cover is the culmination of various processes that include frazil ice growth, ice pan development, juxtaposition and upstream progression taking place. When the ice cover 'bridges' or is present 'bank to bank' across the river channel the increasing frictional resistance causes a rise in the water level. This initial ice cover progression past the gauge is observed as a spike in the water level chart and is depicted as $H_F$ (freeze-over water level) in Fig. 3. The NHP recorded instantaneous water level, up to the minute timing, date and associated daily discharge, as available are manually extracted and given a '0' rating. Instantaneous discharge during ice conditions is not a NHP data product since the open water S-Q relationship is invalid. If no instantaneous record was available, the lower-resolution daily water levels are used to identify the maximum water level occurring after the First B Date with the data quality was rated as '1'. Review of daily meteorological data at proximal climate stations can help the interpretation by knowing that air temperatures remained below 0ºC and the observed spike was not a result of rainfall in the region (Beltaos, 1990). Meteorological data review was accomplished using the 'Search by Proximity' function from: https://climate.weather.gc.ca/historical_data/search_historic_data_e.html. Southern locations generally have a climate station within a 10 km radius; while at some northern locations, it was necessary to assume a representative meteorological site beyond a 200 km radius. The archived hydrometric station analysis (item 7, Sect. 3.1) often includes reference to a nearby meteorological site with: *"Rainfall or temperature records used for estimating the missing periods or the ice affected periods"*. It was generally observed, though not recorded, that freeze-up spikes tend to occur when



temperatures dropped to -10 °C.  While ice jamming at freeze-up is a known occurrence (e.g. Jasek, 1999), there was no
attempt to distinguish these events in the current exercise due to the complex hydrological and hydraulic conditions affecting
these processes. Beltaos (1990) discussed the unlikelihood that a complete ice cover forms at the instant of $H_F$.  A later
recommendation was to define the freeze-up water level as the average water level for one week after formation of a complete
ice cover (Beltaos, 1997).  Following this methodology, the CRID includes all available daily water level at $H_F$ and the
following 29 days for the two following reasons:  (1) allow for calculation of a 7-day average to parameterizes a water level
threshold of exceedance for the ice to detach from channel banks at break-up (Beltaos, 1997) and (2) tabulates water level as
liquid water goes into hydraulic storage and ice formation, temporarily reducing the discharge at the gauge (Prowse and Carter,
2002; Beltaos 2009).

**3.4.2 Ice Cover: $H_{LW1}$, $H_{LQ1}$**

Along with the drainage of surface water storage, a primary source of flow in unregulated rivers during the winter snow and
ice cover period is groundwater. The gradual drawdown of these contributions over the ice cover season leads to a reduction
in river flow with the water level eventually reaching a corresponding minimum value. In small streams, the minimum flow
of the year may occur just after the first extremely cold period (United States Geological Survey, 1977). Since the open water
S-Q relationship does not hold under ice, the NHP daily reported first minimum winter water level ($H_{LW1}$) and estimated first
minimum winter discharge ($H_{LQ1}$) over the ice period may not occur on the same day. For example, Fig. 6 depicts more than
three months of separation between the two on the lower Athabasca River where the higher reported water level in March has
a smaller discharge compared to the November minimum water level event.  The $H_{LQ1}$ is one of several water quality and
aquatic habitat indicators in ice affected rivers (Beltaos and Prowse, 2009; Peters et al., 2014), while an occurrence
synonymous to the first minimum winter water level ($H_{LW1}$) was recently highlighted as a determining factor for navigation
within the Mississippi watershed (Giovando and Daly, 2019).  These data on under-ice minimum magnitude and occurrence
are to inform regional low flow analysis (Beltaos and Prowse, 2009), environmental flow need assessments, water intake
elevations, water withdrawal guidelines and cross-sectional habitat reductions during ice conditions (e.g., Peters et al., 2014).



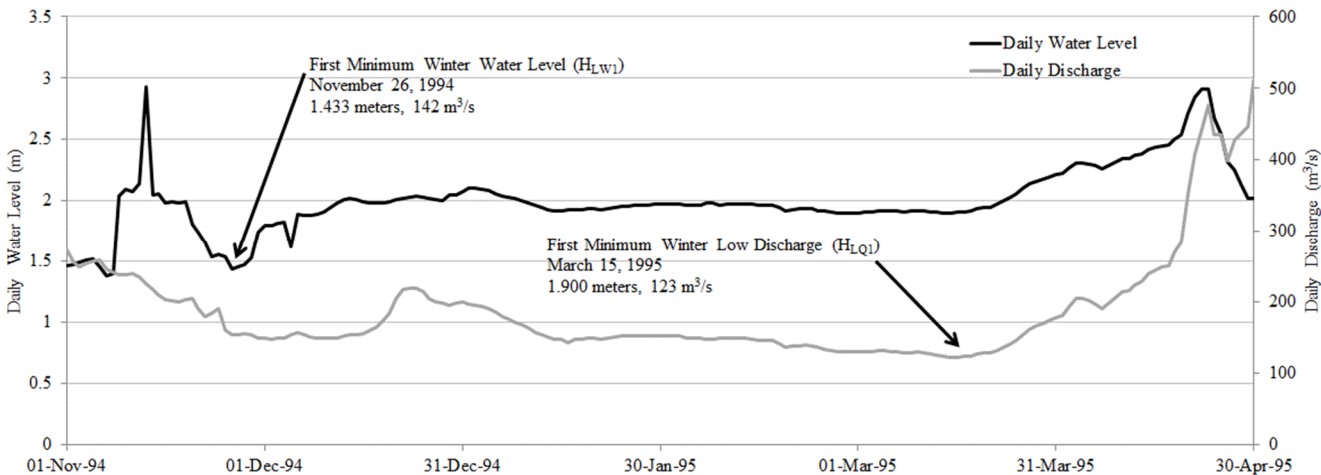

**Figure 6.** Daily reported water level and discharge for the Athabasca River below Fort McMurray (07DA001) for ice affected (B flagged) period spanning November 1, 1994 to April 30, 1995. Note that an increase in water level does not necessarily result in more discharge due to the varying hydraulic effects of ice. Figure adapted from de Rham et al., (2019).

### 3.4.3 Ice Cover: $H_{MWB}$, $H_{MWM}$

Rapidly warming air temperatures (above 0°C) and associated rain on snow events during the ice cover season are the main causes of mid-winter break-up events depicted as water level trace in grey on Fig. 3. These events occur on both regulated (Picco et al., 2003) and unregulated rivers (Newton et al., 2016). The possibility of mid-winter ice jams, elevated water levels risk, and in extreme cases, the freezing of overbank floodwaters as shown in Fig. 7, are major threats to riverside communities and infrastructure (e.g. Beltaos, 2002; Beltaos et al., 2003; Curi et al., 2019). Interpretation of these "winter peaks" from water level records to determine if they are results of ice cover break-up is a challenge (Beltaos, 1990), especially in the absence of other supporting evidence (e.g. site observations, new reports, flood summaries). Similar to freeze-over interpretation (Sect. 3.4.1), the review of daily climate data from nearby stations informs if temperatures exceed 0°C and associated rainfall occurred. During data extraction it was often observed that mid-winter break-up occurrence corresponded with 10's of cm reductions in daily snow on ground for day(s) prior to the event. A review of the discharge measurement summary (item 9, Sect. 3.1) also increased interpretation confidence towards when station visit remarks were available days before or after the "winter peak" alluding to channel ice condition or if discharge measurements were collected from the ice cover or wading.

The instantaneous $H_{MWB}$ represents the onset of ice cover movement at a site during the winter season and is identified as a spike on the rising limb of the water level record. The cause of this spike is a rapid decrease in hydraulic resistance as the ice cover breaks and starts moving downstream. Following the initial break-up event, the water level will typically continue to rise until it reaches a maximum value represented by instantaneous $H_{MWM}$. For some stations, $H_{MWB}$ and $H_{MWM}$ can occur more



than once during a single ice season (e.g. Beltaos, 2002). In such cases, only the first $H_{MWB}$ and the highest $H_{MWM}$ are included
in the CRID. For years with no continuous water level records, daily summaries (item 8, Sect. 3.1) were examined for a
presence of a $H_{MWM}$. NHP notations in the other archival documents (Sect. 3.1) and meteorological data review assisted
judgment on whether these daily maximums likely represented a mid-winter break-up. On occasion, a rudimentary internet
search was used to find alternative verification. Mid-winter break-up sites usually occurred in the temperate zone where B date
flags can be intermittent, leading to complexity and additional interpretation in extracting this variable. For instance, a few
winter break-up events were interpreted to occur during non-B dates because of the extreme water level magnitudes reported.
Closer examination of these events for future studies is recommended.


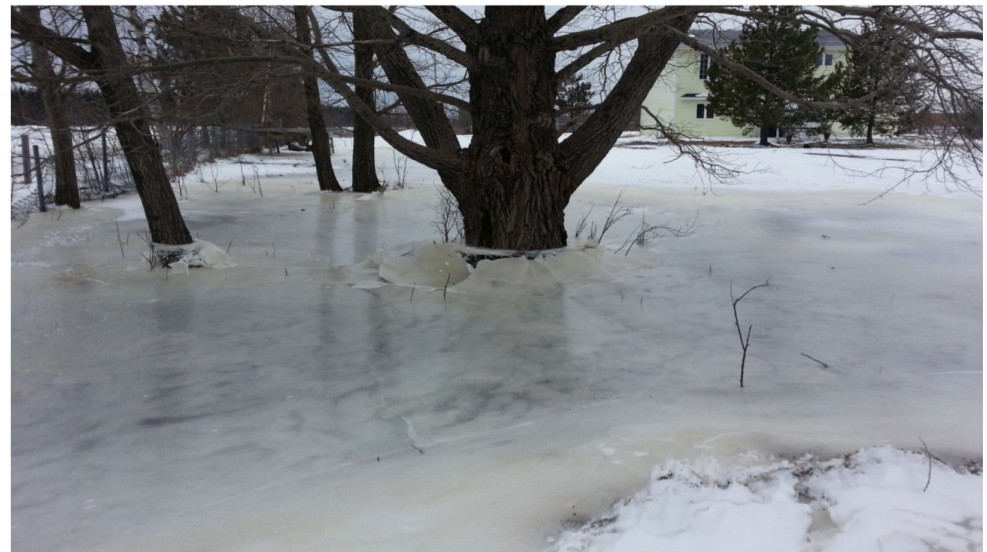


**Figure 7.** Frozen water after mid-winter break-up and over-bank flooding on the Exploits River. Image was taken on February
9, 2013 on Beothuck Street property in Badger, Newfoundland. Ring of frozen ice around the tree trunk indicates the highest
water level. Hydrometric station Exploits River at Badger (02YO013), not a CRID station, is ~ 100 m from this location. Image
from Rebello (2013).

**3.4.4 Ice Cover: $H_{F2}$, $H_{F2\ MAX}$, $H_{LW2}$, $H_{LQ2}$**

The occurrence of ice cover season maximum water levels, not associated with the freeze-up or break-up of the ice cover were
identified from the hydrometric archive and input to the CRID. If there was mid-winter break-up event, an attempt was made
to extract the first of the 7-day maximum average winter water level ($H_{F2}$) after the event. As with $H_F$ (Sect. 3.4.1), these data
mark important parameters for the onset of break-up prediction. No attempt was made to identify an instantaneous $H_{F2}$ since





the CRID archive does not have historical pen recorder charts (Sect. 3.2) much beyond the $H_{MWM}$ event. Examination of more
recent continuous digital water level records reveals that after mid-winter break-up, limited 'stage up', synonymous to $H_F$ was
usually observed. This may be due to the lack of complete ice flush down the channel after $H_{MWM}$. Since large, fragmented ice
blocks likely remain in the channel, the hydraulic resistance and refreezing of the ice cover is probably a less dynamic event.
Daily water level values after mid-winter break-up revealed a pattern of steadily declining daily water levels. If $H_{MWM}$ was
followed by days with no 'B' data flag, $H_{F2}$ was restricted to days when 'B' data flag appear again. As with the first freeze-up
events, $H_{F2}$ and the following 29 days of daily water level were recorded. Water levels within the first 7 days after $H_{F2}$ were
also assessed to extract a maximum ($H_{F2\ MAX}$) daily water level exceeding $H_{F2}$. This variable may more closely match the
instantaneous processes resulting in the $H_F$ occurrence.

Maximum winter water level was also recorded at select locations with no mid-winter break-up event. In this situation, the 7
day average water level beginning at $H_{F2}$ exceeds that commencing of $H_F$. This may correspond with a secondary stage up
during extreme cold events described by (Hamilton, 2003) with Fig. 8 depicting one month between the two peak stages. It is
possible that rising water levels after $H_F$ are caused by secondary consolidation events (Andres, 1999, Andres et al., 2003,
Wazney et al, 2018) however, the daily resolution may be too coarse to capture this short-lived occurrence. An $H_{F2}$ is also
reported (Beltaos, unpublished data) to occasionally occur on the regulated Peace River at Peace Point (07KC001) when mid-
winter flow releases cause increasing water levels but the ice cover remains stable. Some CRID stations reveal 'creeping'
water levels exceeding $H_F$ for most of the ice season (Fig. 9). In such cases, it was not possible to establish $H_{F2}$ and their
occurrences are not included in the CRID. This continuous wintertime increase in water levels could be caused by the
development of anchor ice or continuous build-up of a hanging dam by frazil ice, although both cases require open water at or
upstream of the gauging location. Another possible explanation may be that in the case of Fig. 9, the Pembina drainage area
contains many swamps and muskegs with a water table at or near the surface (Farvolden, 1961)




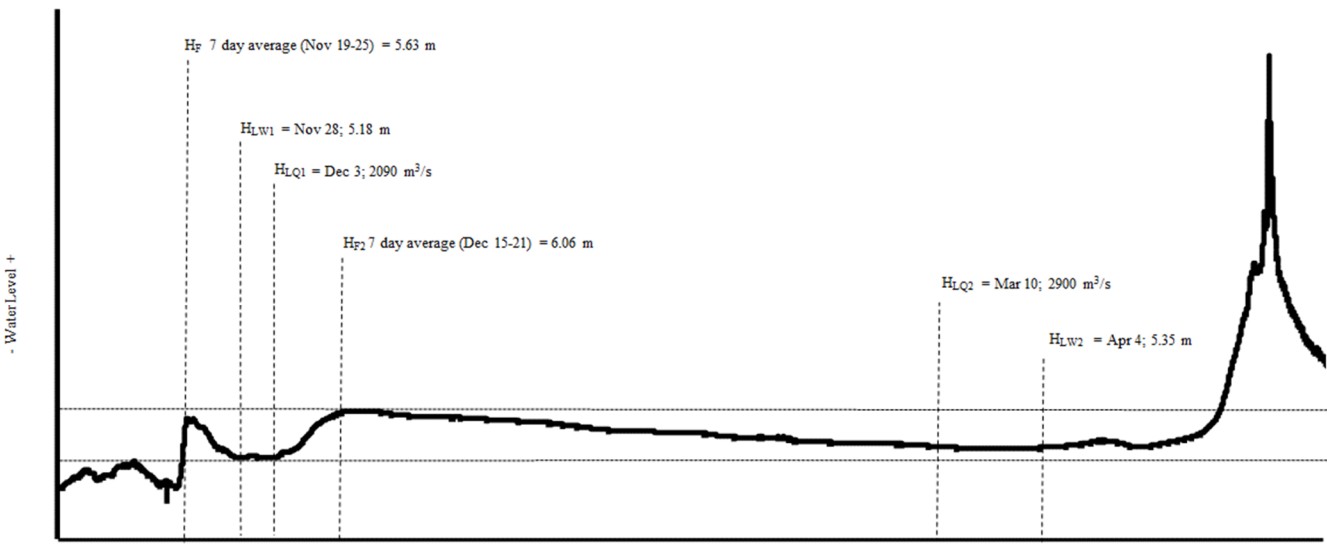


**Figure 8.** Continuous water level record at Mackenzie River at Norman wells during 2010-2011 ice affected flow period. Note
the occurrence of a higher magnitude 7 day average following $H_{F2}$ in comparison to $H_F$ and the corresponding second winter
minimums ($H_{LW2}$ and $H_{LQ2}$) in addition to the first occurrence ($H_{LW1}$ and $H_{LQ1}$)




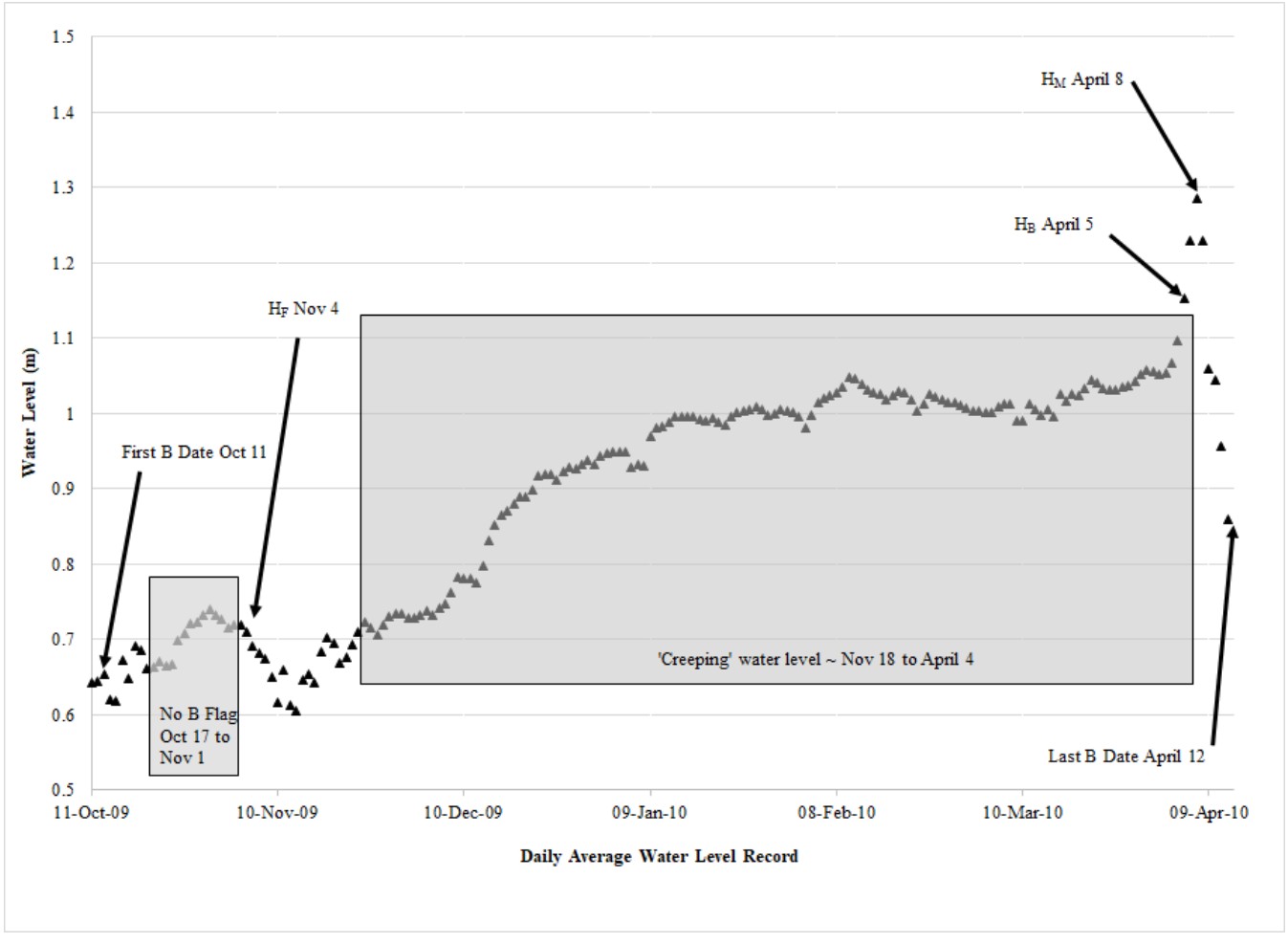

**Figure 9.** Daily water level from First B to Last B date at Pembina River at Jarvie (07BC002) during the 2009-10 ice affected flow season that depict 'Creeping' water level . There are no B data flags from Oct 17 to Nov 1 and daily average water levels 'Creeping' upwards throughout the ice cover period.

Whenever an $H_{F2}$ variable was identified, the ice cover period was examined for a second winter-low water level ($H_{LW2}$) and discharge ($H_{LQ2}$) event. These data were only added to the CRID if $H_{LW1}$ or $H_{LQ1}$ were before $H_{F2}$. At some locations, several months may have lapsed between the first and second occurrences of winter-low events as shown in Fig. 7. The incident of a second winter-low is probably one of the most understudied events in ice covered channels, while it can have all the water quality and navigation related implications as that of the first winter-low events described in Sect. 3.4.2 above.

**3.4.5 Ice Cover: I$_{THICK}$**




Hydrometric technicians visit gauging stations for velocity, water depth, discharge, and water level measurements and
instrument maintenance approximately six to eight times per year, which include both open-water and ice-covered conditions.
During the latter, a measure related to the solid portion of the ice cover thickness is recorded on the site survey note (item 4,
Sect. 3.1). End of ice cover season measurements quantify ice thickness prior to the spring break-up and some cases this may
represent a pre-melt ice thickness, a relevant factor in break-up initiation and potential severity (Beltaos, 1997). Measurements
prior to ~1995 are generally limited to water surface elevation to bottom of ice cover, thus may underestimate the actual
thickness of the ice cover since the density of solid ice is 0.92 that of water and part of the ice cover may float above the water
line depending on the snow loading. Nevertheless, these measurements are assumed to represent the actual ice cover thickness
considering the likely presence of impure ice and snow loads. WSC Regional office and provincial partner protocols for
collection and summary of this ancillary ice thickness data differ, while some of the more recent digital data archives may
have actual ice thickness measurements. Figure 10 shows 19 channel depth and water surface to bottom of ice measurements.
Some hydrometric survey notes report the presence of slush that results in an overestimate of channel ice depth. For the CRID,
all cross-sectional ice thickness measurements were reviewed for the reporting of slush conditions, while all data were plotted
to aid in visual identification and removal of measurements that include slush (see caption for Fig. 10). The remaining
measurements were used to calculate the average river ice thickness ($I_{THICK}$).

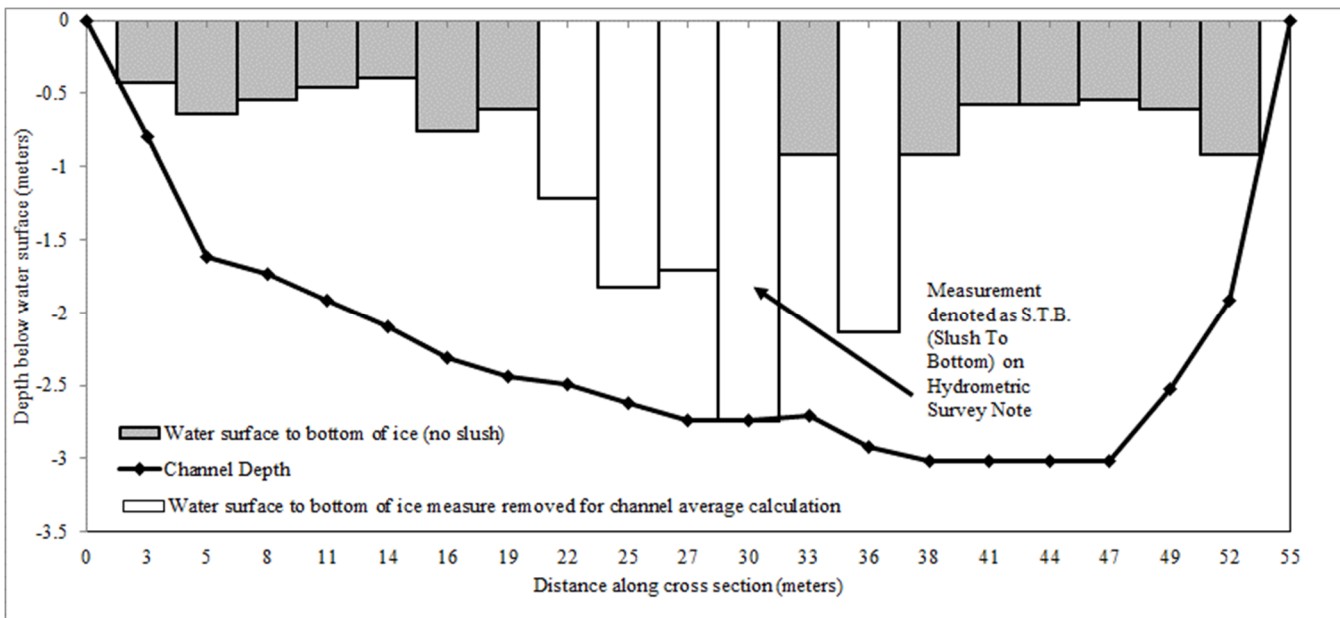



**Figure 10.** A bar plot of the 19 water survey to bottom of ice thickness measurements collected on March 28, 1978 at Nashwaak
River below Durham Bridge (01AL002). The hydrometric survey note indicates measurement at river cross section distance
30 m is S.T.B. (Slush To Bottom). Visual examination of this plot reveals four other measurements (shown with white fill)
which likely include slush. These five measurements are removed when calculating average river ice thickness.

In some years, visits and data collection at hydrometric stations were hampered by weather conditions, logistics or on-ice
safety considerations. As an example, Fig. 10 shows a time series of 47 average ice thickness data points at one CRID location.
Over the time series, the measurement dates range over a 10-week (72 day) time window. In addition to data collection timing,
incomplete archival and scanning for the database may also be a reason for missing or wide ranges in time series. Thus, any
time series analysis of $I_{THICK}$ needs to account for this year-to-year sample date variability. While an attempt was made to
compile the time series of final (season's end) ice thickness measurements, a more detailed climatological analysis will be
required to establish if this measurement was collected prior to the ice cover beginning to melt.

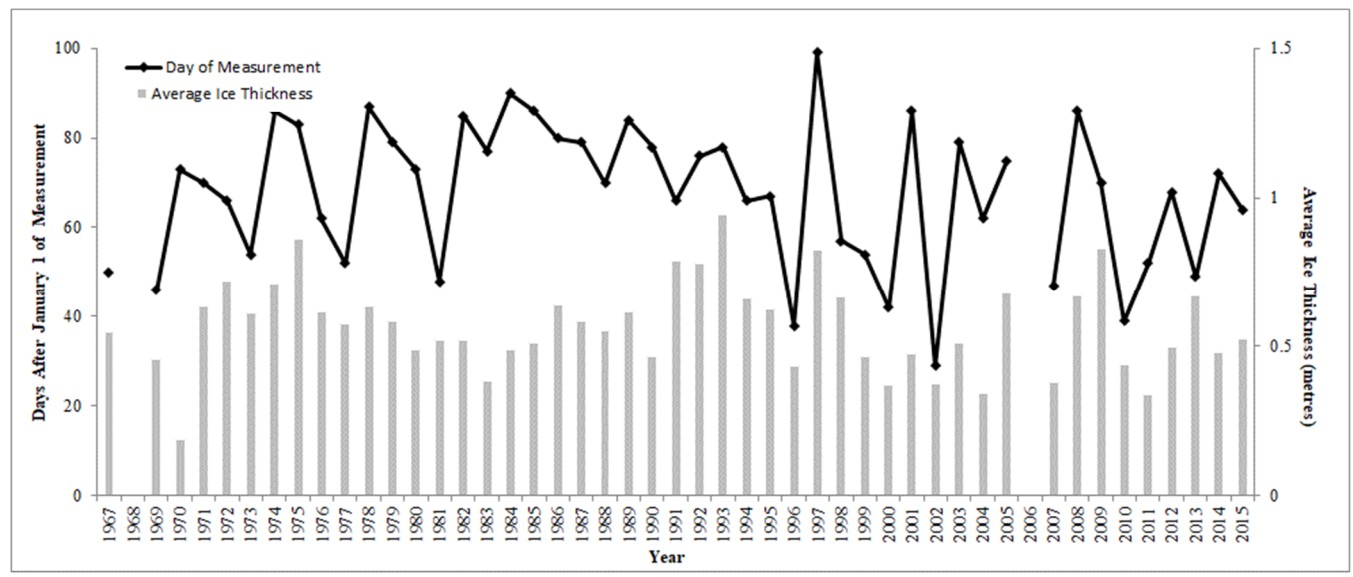



**Figure 11.** Plot showing average ice thickness (grey bars) day of measurement (black line) and at site Nashwaak River below
Durham Bridge (01AL002). Measurement dates input to CRID represent a range of 72 days from a minimum Jan 29 (2002) to
April 9 (1997). Initiation of break-up at this location ranges from Feb 27 (2010) to April 13 (2001) with average of March 25
(84 days after January 1) .

**3.4.6 Break-up: $H_B$, $H_M$, Last B Date**

The end of the river ice season progresses through a continuum of spring break-up initiation ($H_B$), maximum spring break-up
water level ($H_M$) and the last day of ice affected flow (Last B Date). $H_B$ occurs at the initial downstream movement of river
ice cover. The associated decrease in resistance to flow registers as a spike on the rising limb of the water level hydrograph





(see Fig. 3). Beltaos (1990) indicated that identification of break-up initiation can be uncertain and that it is not possible to
establish $H_B$ in the absence of a continuous water level record. Therefore, the timing and magnitude of $H_B$ may be less accurate
than $H_M$, the maximum instantaneous or daily water level established following $H_B$. Data ratings are provided to indicate the
accuracy of these events. The Last B Date was the final day with a B data flag (R data flag for CEHQ sites).

The break-up period can be characterized as either thermal (overmature) or mechanical (dynamic) (Gray and Prowse, 1993;
Beltaos, 2003). In the case of a thermal event, increasing air temperatures and solar radiation inputs during early spring cause
the ice cover to decay. A slow increase in channel flow will prolong the decay period and resulting water levels do not reach
magnitudes much beyond those with similar flow indicated by the open water S-Q relationship. Conversely, a mechanical
break-up is characterized by limited reduction in the mechanical strength of the ice cover and rapid increase in channel flow.
As the rising flow eventually overcomes the resistance of the ice cover, the latter is mobilized in dynamic fashion and quickly
breaks down into slabs and blocks, which eventually are arrested by still-intact ice cover to form ice jams, typically at
morphologically conducive locations such as constrictions and abrupt slope reductions. Earlier analysis reports indicated that
$H_M$ can far exceed water levels that occur under similar open-water flow conditions (de Rham et al., 2008a; von de Wall et al,
2009, 2010; von de Wall, 2011). Depending on their location and persistence, ice jams lodged at or below the gauge site affect
the local water levels to a varying degree. Jams lodged upstream of the gauge only affect the local water level upon their
release, which generates a sharp wave (called jave for short, Beltaos, 2013). A jave is yet another dynamic mechanism that
can generate the identified $H_M$ water level. Highly dynamic events, initiated with minimal or negligible ice cover decay, are
sometimes referred to as "premature" and typically result from mid-winter thaws accompanied by intense rain-on-snow runoff
events (Deslauriers, 1968). It is likely that much of the CRID mid-winter data described above in Sect. 3.4.3 are these highly
dynamic events. The less common "overmature" break-up sequence was observed at some CRID stations with no obvious
"spiking" of water levels. An example water level of this occurrence on the Peace River in 1982 (Fonstad, 1982) is included
in Beltaos (1990).

Figure 12 shows an example timeline, with images of changing ice conditions for the year 2010 break-up sequence at Hay
River near Hay River (07OB001). Unfortunately, images at the extracted CRID timings of $H_B$ and $H_M$ are not available;
however, images 5 minutes later are illustrative: The night time image (April 24, 4:30) shows a large chuck of ice along the
left channel bank indicating fracture of the ice cover and initiation of break-up. One hour later, the near open channel condition
(April 24, 5:30) highlights the downstream forces involved in flushing of in-channel ice. The image on April 25 at 15:30 shows
stranded ice fragments on the channel banks, 5 minutes after $H_M$ (April 25, 15:25). The peak water levels at $H_M$ and subsequent
water level drop would raft and settle the ice fragments outside the channel. While no Last B Date image is available, it is
notable that the river ice break-up processes described occur prior to this date. While spring break-up peak water level
magnitude and timing in the CRID have high degree of accuracy, classification of events as ice jam or not, was not pursued as
this would require local observations and/or photos. The last B date is sometimes used to represent break-up for time series
analysis (e.g. Zhang et al., 2001; Chen and She, 2019) and a recent publication used B dates and discharge to assess trends in
ice jam flooding events (Rokaya et al., 2018). Unlike using the last B date as a surrogate and/or index, the water-level based
data in the CRID provides the science community with a direct and thus more accurate data set towards analysis of spring
break-up timing, magnitude and processes. For instance, the identification of $H_M$ provides the means to assess change in the
flow magnitude driving spring breakup flooding, which would not be possible with discharge analysis alone and/or solely
identifying the last B date.

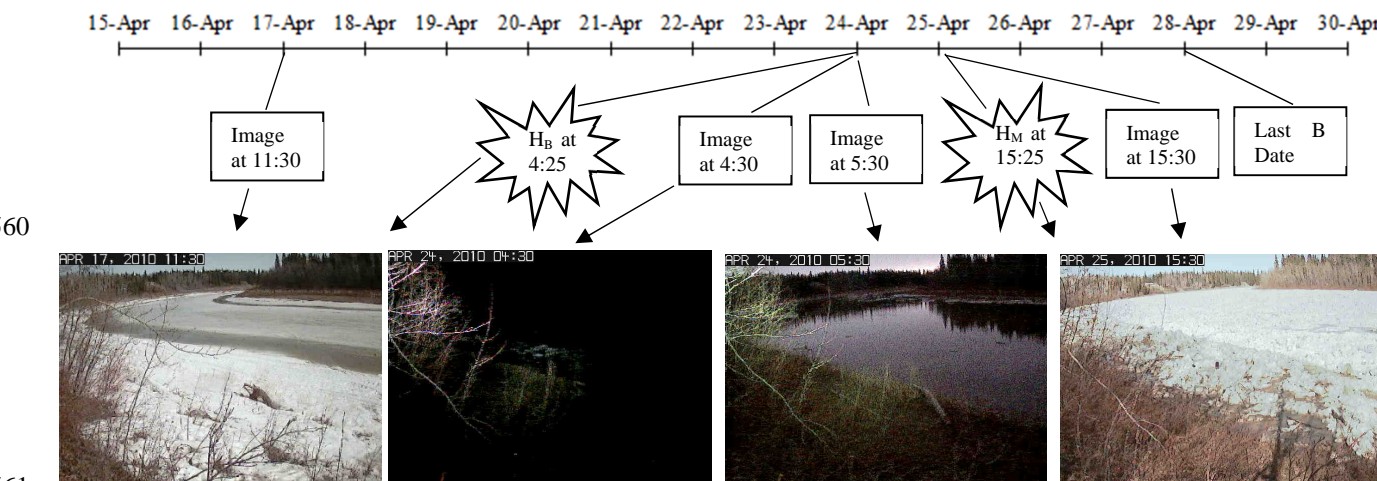


**Figure 12.** Left: Image looking upstream taken 7 days prior to spring break-up initiation ($H_B$) of April 24, 2010, 04:25 at
location Hay River near Hay River (07OB001). Channel width of approximately 63 meters. Centre, left is a night time image
5 minutes after $H_B$ and shows evidence of fragmented ice in the channel. Centre, right is 65 minutes after $H_B$ and shows channel
nearly clear of ice. Right image is 5 minutes after maximum spring break-up water level on April 25, 2010, 15:25. Stranded
ice on channel banks indicates higher water levels. Last B date was April 28, 2010. Images courtesy of University of Alberta
River Ice Research Group.

**3.4.7 Open-Water: $H_O$**

The CRID includes the magnitude and timing of the maximum open-water level ($H_O$) and the associated discharge value at
each station along with data quality rating. These data are extracted from the hydrometric archives and are easily verified as
NHP web pages generally report both daily and instantaneous maximum annual discharge and timing. In the event of damaged
or non-functioning instrumentation, NHP or CEHQ may estimate (data flagged with E) daily discharge values. The S-Q
relationship (Sect. 3.1) can be used to calculate the associated water level. Gerard and Karpuk (1979) provided one of the
earlier examples of comparing maximum ice affected to open water levels on the Peace River. These types of analysis inform
the hydrological community on the importance of looking at ice effects as the likely causes of maximum annual flood levels



(e.g. de Rham et al., 2008a). Visual examination of $H_O$ time series on a stage-discharge plot is a cursory method to identify
station movements, benchmark or datum shifts, or changes to the stage-discharge relationship. This is discussed in more detail
below.

**3.5 Data Accuracy and Precision, Quality Control and Interpretation**

The accuracy and precision of extracting water level, discharge and timing of the CRID variables was as follows. For the five
grey shaded instantaneous variables in Table 2 ($H_F$, $H_{MWB}$, $H_{MWM}$, $H_B$, $H_M$, $H_O$), extraction precision of up to 2 decimals for
the pre-1978 (data in feet) and 3 decimals for the post-1978 (data in meters) was possible based on visual inspection of the
continuous (i.e. analog) water level recording charts (pre ~ 2000). All imperial data in feet were converted to metres using
factor of 0.3048 and are reported to 3 decimals in the CRID database. Although much of the water level records are continuous,
the visual extraction method often limited the associated timing of an event to a 15-minute resolution. Instantaneous timing at
finer resolution within the CRID were usually obtained from alternative archival documents (e.g. Annual Water Level Page,
Station Analysis or published online summaries). The wide-spread use of digital water level recording instrumentation after
the year ca. 2000 decreased the temporal resolution (i.e., accuracy) of water level records as data collection interval varied
from 5 to 15 to 60 minutes. Some data loggers also recorded hourly to sub-hourly maximum and minimums, which increased
the accuracy towards instantaneous events, though selection does require judgement.

Quality Control (QC) for the CRID has included preliminary data analysis and peer review of associated publications (Table
1). CRID station data were initially compiled as single station Excel files which include all extracted water level, discharge,
date and time and accuracy rating, average ice thickness along with time series plots for visual identification of outliers. A
separate station Excel file contains all available ice thickness measurements and averages calculation. All finalized station
data were compiled in to a single .csv file (118 columns x 22,736 rows with 464,891 cell entries) for further QC. This single
spreadsheet was examined for data entry errors using the filter and count capabilities inherent to Excel.

A quantification of data interpretation and input errors was undertaken. Automated scripts were used to extract CRID
associated daily discharge values along with First and Last B date from a bulk download of all available NHP daily flow data.
Discharge values input to the CRID were found to be between 4.7% to 7.8% depending on the variable. Mid-winter associated
events had the highest input error at 16%. For ice seasons when both a First and Last B Date were available, an input error of
7.5% was found. All erroneous daily discharge and B Date values were replaced. The CRID initiation of break-up ($H_B$) time
series at site Red River near Lockport (05OJ010) was provided to Becket (2020) who reported: of the 34 years, 3 years of
timing were revised based on evidence in newspapers (an ancillary evidence source not included in the CRID), while 2 years
were found to be incorrectly interpreted and input to the CRID. One year was 12:00 hours too early and one year 2 days too
early. Based on these QC activities the CRID likely has a 5-10% data interpretation/entry error. While it would be impractical



to review the entire database for errors, users are encouraged to undertake their own QC activities and review the data
disclaimer in Sect. 7. Original archival documents can be requested from the authors. Upload of this archive to a more
convenient format may be pursued in the future.

Extraction of river ice data from hydrometric records is a time consuming and detail-oriented task. The average time needed
by an experienced investigator to identify and input data associated with the 15 CRID variables for a one-year period at a
single station was about 1 hour . Besides the laborious nature of this work, additional uncertainties are caused by site-specific
phenomena that can have varying effects on water level. The NHP archives include field observations of beaver dam in channel,
open water leads at, upstream or downstream of the gauge, percentage of ice cover at gauge, water flowing between the ice
layers and anchor ice at a cross section. While these types of observations are not part the CRID, users should be aware of
such factors that add further complexity to wintertime water level interpretation. Furthermore, collection of data using a stilling
well (von de Wall, 2011) also could affect resultant water level interpretation. Since river ice processes can be site specific
users should be aware of possible spatial discrepancy in location of gauge site versus where ice thickness and flow
measurements are collected.  Access to ice cover and worker safety are field based considerations which can result in a
wintertime cross section measurements taken meters or kilometres  upstream or downstream from the actual gauge. Another
consideration is that many gauges are located near a bridge, which provides a safe platform from which water velocity
measurements can be performed. Bridge pilings would change the hydraulics and very likely the ice condition on a river
channel. Finally, changes to watershed characteristics such as urbanization and agriculture likely have effects on river ice
hydrology.

CRID users should also bear in mind that all variables were transcribed directly as recorded in the NHP archive. There is no
tabulation of: at-station movements, benchmark or datum shifts, or changes to the stage-discharge relationship. Since river ice
processes are site specific, time series analysis of phenology or water level data needs to account for these three factors.  For
example, Fig. 12 shows all Albany River CRID data on a stage-discharge plot.  The WSC website informs that the station was
relocated in 1988 with a new gauge height, and as a result the maximum open-water level magnitudes (blue circles) plots as
two separate populations which are not directly comparable for many types of analysis.



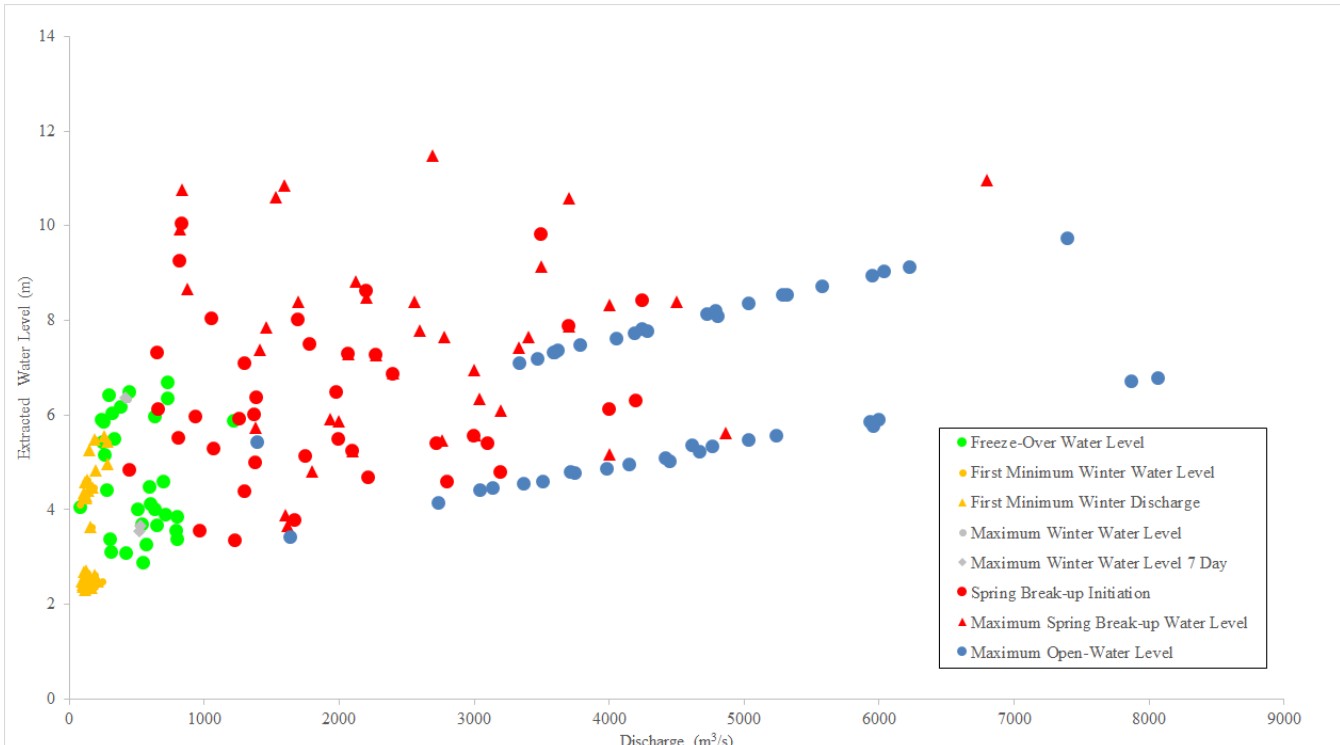

**Figure 13.** A stage-discharge plot of Canadian River Ice Database variables for site Albany River near Hat Island (04HA001). Time series for Maximum Open-Water Level (blue circles) plot as 2 separate populations. This gauge was relocated approximately 3.5 km downstream on Sept 29, 1988.

## 4  Discussion

### 4.1 The CRID

A two-decade effort has culminated in the CRID which covers a network of 196 hydrometric stations with data up to Dec 31, 2015 that represent 10,378 station-years of active operation. During the first decade, the work focused primarily on the spring break-up season, while for the past decade it was expanded to include the entire period of ice-affected flow. The 15 variables occur at different stages of the season (Table 4) and include minimum daily and maximum instantaneous water level events, ice thickness along with discharge-based metrics and provide a comprehensive baseline dataset for research purposes.  The CRID is available for download at: https://doi.org/10.18164/c21e1852-ba8e-44af-bc13-48eeedfcf2f4 (de Rham et al., 2020)

In total, the CRID holds 72,595 variables with more than 460,000 data entries of water level, discharge, date, time and data quality rating based on the review of over 100,000 hydrometric archive files. Tabulation of the 6,094 ice thickness





measurements required examination on the order of 100,000 cross-sectional measurements and removal of slush affected data.
In terms of data completeness, extraction of maximum open-water level ($H_o$) was the most successful covering 9,705 (94%)
of the 10,378 active station years.  Similarly, the 8,933 (9,240) first (last) day with backwater due to ice and 8,178 first
minimum winter discharge populate the majority of active station-years and attest to the NHP historical mandate to publish
discharge information. Freeze-over water level and maximum spring break-up water level were extracted from 72% and 80%
of those years reporting first and last B date. This first known attempt to centralize data on mid-winter break-up occurrence
includes 467 maximum mid-winter break-up water level and 362 associated mid-winter break-up initiation events.  The data
quality rating presented in Table 4 confirms that the NHP archives is a high quality source of river ice information with 82%
of data rated as '0'. Although some of the data have lower quality ratings, their inclusion increases the population size and
helps provide a more complete spatial and temporal coverage over Canada.

While the CRID represents the largest existing effort to extract river ice variables from hydrometric archives, it does not
provide a complete time series of ice events at the near 2,800 active and 5,500 discontinued hydrometric stations in Canada.
However, it covers a representative sample with six station types (Table 4), including natural and regulated sites along with
their status as active, or discontinued during time of operation up until Dec 31, 2015. Regulated locations are also split into
homogeneous and heterogeneous depending on when the regulation began during the measurement timeline. Active stations
data comprise over 90% of the CRID.  Discontinued stations provide additional information and help increase the density of
the network. Reasons for less than complete at-station time series include seasonal operation, damage to water level recording
instrumentation, no available hydrometric archive for particular year, or loss of information during the CRID archival and
scanning process.

**Table 4.** Total number of variables that populate the Canadian River Ice Database and their Data Quality Ratings. Grey shading
indicates an attempt was made to extract the instantaneous water level.  Also included are column totals per river type:
Natural/Regulated, Active/Discontinued, Homogeneous/Heterogeneous.





| Season | Variable | Total Number of Variables | Data Quality Rating | | | Natural | | Regulated | | | |
|---|---|---|---|---|---|---|---|---|---|---|---|
| | | | 0 | 1 | 2 | Active | Discontinued | Homogeneous, Active | Homogeneous, Discontinued | Heterogeneous, Active | Heterogeneous, Discontinued |
| Freeze-up | First Day With Backwater Due To Ice | 8,933 | no Data Quality rating | | | 5,754 | 806 | 1,204 | 130 | 1,022 | 16 |
| Freeze-up | Freeze-Over Water Level | 6,547 | 4,794 | 1,592 | 161 | 4,142 | 466 | 949 | 106 | 881 | 3 |
| Ice cover | First Minimum Winter Water Level | 4,767 | 4,557 | 193 | 17 | 2,861 | 214 | 823 | 103 | 766 | 0 |
| Ice cover | First Minimum Winter Discharge | 8,178 | 8,114 | 62 | 2 | 5,301 | 764 | 1,077 | 111 | 925 | 0 |
| Ice cover | Mid-Winter Break-Up Initiation | 362 | 359 | 3 | 0 | 249 | 11 | 54 | 8 | 40 | 0 |
| Ice cover | Maximum Mid-Winter Break-Up Water Level | 467 | 392 | 70 | 5 | 308 | 22 | 77 | 9 | 51 | 0 |
| Ice cover | Maximum Winter Water Level | 1,954 | 1,816 | 39 | 99 | 1,180 | 104 | 329 | 16 | 325 | 0 |
| Ice cover | Maximum Winter Water Level 7 Day | 1,952 | 1,849 | 27 | 78 | 1,180 | 104 | 329 | 16 | 325 | 0 |
| Ice cover | Second Minumum Winter Water Level | 798 | 794 | 4 | 0 | 407 | 39 | 186 | 7 | 159 | 0 |
| Ice cover | Second Minimum Winter Discharge | 709 | 709 | 0 | 0 | 325 | 37 | 172 | 4 | 171 | 0 |
| Ice cover | River Ice Thickness | 6,094 | no Data Quality rating | | | 4,163 | 416 | 762 | 59 | 669 | 25 |
| Break-up | Spring Break-Up Initiation | 5,534 | 5,070 | 333 | 131 | 3,541 | 323 | 885 | 121 | 641 | 23 |
| Break-up | Maximum Spring Break -Up Water Level | 7,355 | 5,428 | 1,571 | 356 | 4,483 | 503 | 1,216 | 168 | 914 | 44 |
| Break-up | Last Day With Backwater Due To Ice | 9,240 | no Data Quality rating | | | 5,816 | 788 | 1,380 | 186 | 1,024 | 46 |
| Open-Water | Maximum Open-Water Level | 9,705 | 5,705 | 3,728 | 271 | 6,121 | 826 | 1,408 | 184 | 1,119 | 47 |
| | Column Total: | 72,595 | 39,587 | 7,622 | 1,122 | 45,831 | 5,423 | 10,851 | 1,228 | 9,032 | 204 |

## 4.2 Utility of the Database and Research Needs

The CRID can be used for the study of river ice processes and the key characteristics of different ice regimes that are encountered within Canada and how these characteristics may be changing over time. From a practical standpoint, there are many flood-prone sites across Canada, and various municipalities often commission engineering studies to assess open-water and ice-jam flood risk. If a site happens to be included in the database, much effort could be saved by, for example, having a ready historical record of maximum ice-influenced levels and related flows, their time of occurrence, and the thickness of the winter ice cover. Maximum ice affected water levels in the CRID are a good candidate to populate the National Ice Jam Database (Muise et al., 2019), a Natural Resources Canada contribution to the Federal Floodplain Mapping Guidelines (https://www.publicsafety.gc.ca/cnt/mrgnc-mngmnt/dsstr-prvntn-mtgtn/ndmp/fldpln-mppng-en.aspx).

It has been established that extreme flooding in Canadian rivers is very often the result of ice jams, with water levels exceeding those occurring under open-water conditions at much higher discharges (e.g. Gerard, 1989). However, river ice is generally omitted from major Canadian hydrological and hydraulics research initiatives (eg. NSERC FloodNet, 2015), likely as a result of the limited field data representing these complex and sometimes chaotic events of ice formation, growth and decay. Many national-scale assessments of flooding make little mention of river ice conditions, their implications to extreme water levels and the inherent challenges encountered in the estimation and reporting of discharge under ice (e.g., Cunderlink and Ouarda; 2009; Burn and Whitfield, 2016). Variables from the CRID could likely be incorporated in future hydrological initiatives and flood assessments.





Some classification schemes have been proposed to help educate current and future hydrological practitioners on the types and
significance of river ice processes and ice jams (IAHR Working Group on River Ice Hydraulics 1986; Turcotte and Morse,
2013). Beltaos and Prowse (2009) also made numerous research recommendations towards the study of river ice conditions.
Examples include: calculation of trends in the frequency and magnitude of ice jams and thickness and strength of pre break-
up ice covers and evaluation of climate-induced changes on river ice hydrology and quantification of intervals between major
river ice events. The CRID provides the necessary baseline data for a complete national assessment of river ice conditions and
can help identify rivers/regions where climate change adaptation may be of high priority.
There are a variety of other research questions that can be addressed using the CRID. Many were detailed in CRIPE 2019
proceedings (de Rham et al., 2019) and are reiterated/updated here: application of site-specific break-up forecast
methodologies (e.g., Beltaos, 1997; Beltaos et al., 2003); flood studies (Buttle et al., 2016); evaluation of locations using the
global river ice classification model (Turcotte and Morse 2013); cold-regions ecological assessments (e.g. Peters et al., 2014;
2016); baseline information for under-ice sediment transportation studies (as reviewed by Turcotte et al., 2011) and riverine
habitats stressors (as reviewed by Prowse and Culp 2008); calibration and validation of river ice hydrology (Morales-Marin et
al., 2019) and hydraulics (Lindenschmidt, 2017) modelling efforts; and ground truth observations for remote sensing
applications (Pavelsky and Smith 2004; Yang et al., 2020).
**5 Conclusion**
The Watershed Hydrology and Ecology Research Division of Environment and Climate Change Canada has compiled the
CRID for public access through the Government of Canada open data portal. This effort follows the recommendation of the
1990 CRIPE sponsored report *Working Group on River Ice Jams*, specifically *Chapter 2: Guideline for Extraction of Ice-*
*Break-Up Data From Hydrometric Station Records* (Beltaos, 1990). National Hydrometric Program gauge records proved to
be very valuable sources of field data for parameterization of ice related hydrologic events on Canadian rivers. This work
involved reviewing over 10,000 station years of data from a network of 196 stations, covering a range of stream types and
climatic regions, to identify and extract recorded data corresponding to 15 variables comprising water levels, discharges,
timings, ice thickness, and data quality ratings. Close to 73,000 records of river ice variables are now available to the water
research community. While many research avenues are possible, it is recommended that periodic updates be made to this
database since a longer time series record is of more value. It is fortunate that much of the data acquisition tasks, discussed
above could be automated using the Aquarius platform currently in use by NHP partner organizations (S. Hamilton, pers.
comm). It is also recommended that a tabulation of station movements, benchmark or datum shifts, and changes to the stage-
discharge relationship be compiled to rectify the site-specific nature of river ice conditions. Lastly, the CRID follows on several
other notable national and international efforts to compile river ice information. The Global Lake and River Ice Phenology
Database (Benson et al., 2000), the Canadian Ice Database (Lenormand et al., 2002), CRREL Ice Jam Database (Carr et al.,



2015), and Russian River Ice Thickness and Duration database (updated by Shiklomanov and Lammers, 2014) represent major
open data contributions to river ice science over the past two decades. The CRID expands on the number of variables
considered, as well as, the temporal and spatial scope of these earlier databases for stations in Canada. The work highlights the
excellence of NHP agencies in the collection and dissemination of hydrometric data, adds value to the NHP archive and
delivers on Environment and Climate Change Canada's commitment to making water science knowledge and data openly
available to the scientific community and the general public.

**6 Data Availability**

The CRID is available for download as a single .csv format file on the Government of Canada Open Data portal at:
https://doi.org/10.18164/c21e1852-ba8e-44af-bc13-48eeedfcf2f4 (de Rham et al., 2020). A 0.5 Terabyte digital archive of all
available scanned and digital hydrometric archives contains around 30,000 folders and over 100,000 files is stored on ECCC
server. This archive is available up request.

**7 Data Disclaimer**

Environment and Climate Change Canada employs every reasonable effort whenever feasible, to ensure the currency, accuracy
and precision of the information provided. However, there are some limitations due to the sources of the data and the
technology used in its processing and management. Furthermore, the material or any data derived using the data is subject to
interpretation. Users are responsible for verifying that the supplied material is appropriate for the use or application for which
they wish to employ it.

**8 Acknowledgements**

The authors are extremely grateful to NHP partner organizations (WSC, CEHQ, government of AB and SK) along with the
regional staff for providing access to the hydrometric data archives, in-kind support and technical input though multiple phases
of this long-term project. We thank the effort of hydrometric field workers in the collection and maintenance of this data over
the period covered in the CRID. The authors want to make mention to individuals (with affiliation at the time) who spent many
hours compiling and extracting data at various phases of the study: Tom Carter, Martin Lacroix, Jennifer Pesklevits (ECCC),
Dwayne Keir, Kyle Eyvindson, Shannon Croutch (University of Saskatchewan), Steeve Deschenes, Jane Drengson, Holly
Goulding, Graham McGrenere, Peter Bi, Kirsten Brown, Simon von de Wall (University of Victoria). Thank-you to Josh
Hartmann (University of Victoria) for his work automating the compilation of the CRID from individual Excel files and quality
control of the B dates and discharge values.



**9 Author Roles**

LD coordinated this study completing data extraction, data entry, and quality control and wrote this manuscript (MS). YD
supervised this study as PI since 2017 and reviewed the MS. SB conceptualized extraction of river ice related data from
hydrometric records in 1990, provided technical guidance throughout the study and reviewed the MS. DP provided technical
input towards data extraction, data quality, ecological and flood aspects and reviewed the MS.  BB advised on river regulation,
hydroclimatic regions and reviewed the MS. TP, ECCC Emeritus Scientist since 2017, initiated this study as a PI in the late
1990s.

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





**Appendix**

**Table A1:** List of the 196 National Hydrometric Program stations which comprise the Canadian River Ice Database. Data extraction time period are shown in column 'Start' and 'End'. Location with (RIVIERE) in water course are in Quebec. Column 'Type' is the regime as Natural, Active (NA); Natural, Discontinued (ND); Regulated, Heterogeneous, Active (RHEA); Regulated, Heterogeneous, Discontinued (RHED); Regulated, Homogeneous, Active (RHOA) Regulated, Homogeneous Discontinued (RHOD)

| Station Number | Start | End | Type | Water Course | Station Number | Start | End | Type | Water Course | Station Number | Start | End | Type | Water Course |
|---|---|---|---|---|---|---|---|---|---|---|---|---|---|---|
| 01AL002 | 1961 | 2015 | NA | NASHWAAK RIVER AT DURHAM BRIDGE | 04JG001 | 1966 | 2015 | RHOA | KENOGAMI RIVER NEAR MAMMAMATTAWA | 07NB001 | 1921 | 2015 | RHEA | SLAVE RIVER AT FITZGERALD (ALBERTA) |
| 01AN002 | 1974 | 2015 | NA | SALMON RIVER AT CASTAWAY | 04LD001 | 1920 | 2015 | RHOA | GROUNDHOG RIVER AT FAUQUIER | 07OB001 | 1921 | 2015 | NA | HAY RIVER NEAR HAY RIVER |
| 01AP004 | 1961 | 2015 | NA | KENNEBECASIS RIVER AT APOHAQUI | 04LG002 | 1959 | 1982 | RHOD | MOOSE RIVER AT MOOSE RIVER | 07OB003 | 1974 | 2015 | NA | HAY RIVER NEAR MEANDER RIVER |
| 01BC001 | 1962 | 2015 | NA | RESTIGOUCHE RIVER BELOW KEDGWICK RIVER | 04LJ001 | 1959 | 2015 | NA | MISSINAIBI RIVER AT MATTICE | 07OC001 | 1969 | 2015 | NA | CHINCHAGA RIVER NEAR HIGH LEVEL |
| 01BH005 | 1970 | 2015 | NA | DARTMOUTH (RIVIERE) EN AMONT DU RUISSEAU DU PAS DE DAME | 04LM001 | 1972 | 2015 | NA | MISSINAIBI RIVER BELOW WABOOSE RIVER | 07PA001 | 1968 | 2015 | NA | BUFFALO RIVER AT HIGHWAY NO. 5 |
| 01BO001 | 1918 | 2015 | NA | SOUTHWEST MIRAMICHI RIVER AT BLACKVILLE | 04ME003 | 1959 | 2015 | RHEA | ABITIBI RIVER AT ONAKAWANA | 08AB001 | 1974 | 2015 | NA | ALSEK RIVER ABOVE BATES RIVER |
| 01BP001 | 1951 | 2015 | NA | LITTLE SOUTHWEST MIRAMICHI RIVER AT LYTTLETON | 04NA001 | 1924 | 2015 | NA | HARRICANA (RIVIERE) 3,1 KM EN AVAL DU PONT-ROUTE 111 A AMOS | 08CE001 | 1954 | 2015 | NA | STIKINE RIVER AT TELEGRAPH CREEK |
| 01BQ001 | 1961 | 2015 | NA | NORTHWEST MIRAMICHI RIVER AT TROUT BROOK | 04NB001 | 1967 | 2004 | ND | TURGEON (RIVIERE) EN AMONT DE LA RIVIERE HARRICANA | 08CF001 | 1971 | 1995 | ND | STIKINE RIVER ABOVE BUTTERFLY CREEK |
| 01BV006 | 1964 | 2015 | NA | POINT WOLFE RIVER AT FUNDY NATIONAL PARK | 05AA023 | 1949 | 2008 | ND | OLDMAN RIVER NEAR WALDRON'S CORNER | 08EE004 | 1930 | 2015 | NA | BULKLEY RIVER AT QUICK |
| 02EC002 | 1913 | 2015 | NA | BLACK RIVER NEAR WASHAGO | 05AB021 | 1908 | 2015 | RHEA | WILLOW CREEK NEAR CLARESHOLM | 08JC001 | 1915 | 2015 | RHEA | NECHAKO RIVER AT VANDERHOOF |
| 02FC001 | 1911 | 2015 | RHOA | SAUGEEN RIVER NEAR PORT ELGIN | 05AC003 | 1918 | 2015 | RHOA | LITTLE BOW RIVER AT CARMANGAY | 08JC002 | 1950 | 2015 | RHEA | NECHAKO RIVER AT ISLE PIERRE |
| 02GA014 | 1947 | 2015 | RHEA | GRAND RIVER NEAR MARSVILLE | 05AD028 | 1966 | 2015 | RHOA | WATERTON RIVER NEAR GLENWOOD | 08KB001 | 1950 | 2015 | NA | FRASER RIVER AT SHELLEY |
| 02GA034 | 1967 | 2015 | RHOA | GRAND RIVER AT WEST MONTROSE | 05BJ001 | 1894 | 2015 | RHEA | ELBOW RIVER BELOW GLENMORE DAM | 08KH006 | 1939 | 2015 | NA | QUESNEL RIVER NEAR QUESNEL |
| 02GB001 | 1912 | 2015 | RHEA | GRAND RIVER AT BRANTFORD | 05BJ004 | 1923 | 2015 | NA | ELBOW RIVER AT BRAGG CREEK | 08LF051 | 1951 | 2015 | NA | THOMPSON RIVER NEAR SPENCES BRIDGE |
| 02GD021 | 1978 | 2015 | NA | THAMES RIVER AT INNERKIP | 05BL024 | 1970 | 2015 | RHOA | HIGHWOOD RIVER NEAR THE MOUTH | 08LG007 | 1911 | 2009 | RHOD | NICOLA RIVER NEAR MERRITT |
| 02HL005 | 1965 | 2015 | NA | MOIRA RIVER NEAR DELORO | 05CB001 | 1960 | 2015 | NA | LITTLE RED DEER RIVER NEAR THE MOUTH | 08LG010 | 1911 | 2015 | RHOA | COLDWATER RIVER AT MERRITT |
| 02LG005 | 1972 | 2015 | NA | GATINEAU (RIVIERE) AUX RAPIDES CEIZUR | 05CC001 | 1912 | 2015 | NA | BLINDMAN RIVER NEAR BLACKFALDS | 08LG048 | 1965 | 2015 | NA | COLDWATER RIVER NEAR BROOKMERE |
| 02LH004 | 1926 | 2005 | ND | PICANOC (RIVIERE) PRES DE WRIGHT | 05CC007 | 1962 | 2015 | NA | MEDICINE RIVER NEAR ECKVILLE | 08LG049 | 1915 | 2015 | RHOA | NICOLA RIVER ABOVE NICOLA LAKE |
| 02NE011 | 1965 | 2015 | NA | CROCHE (RIVIERE) À 2,6 KM EN AVAL DU RUISSEAU CHANGY | 05FF001 | 1911 | 1994 | RHED | BATTLE RIVER AT BATTLEFORD | 08MB005 | 1970 | 2015 | NA | CHILCOTIN RIVER BELOW BIG CREEK |
| 02NF003 | 1931 | 2015 | NA | MATAWIN (RIVIERE) A SAINT-MICHEL-DES-SAINTS | 05AE007 | 1944 | 1994 | RHOD | EYEHILL CREEK NEAR MACKLIN | 08NB005 | 1944 | 2015 | NA | COLUMBIA RIVER AT DONALD |
| 02OA054 | 1970 | 2015 | RHOA | CHATEAUGUAY (RIVIERE) À 2 KM EN AMONT DU PONT-ROUTE 132 | 05GC006 | 1962 | 2015 | RHOA | EAGLE CREEK NEAR ENVIRON | 08NL007 | 1914 | 2015 | NA | SIMILKAMEEN RIVER AT PRINCETON |
| 02OE027 | 1956 | 2015 | NA | EATON (RIVIERE) PRES DE LA RIVIERE SAINT-FRANCOIS-3 | 05GG001 | 1910 | 2015 | RHEA | NORTH SASKATCHEWAN RIVER AT PRINCE ALBERT | 08NL038 | 1914 | 2015 | NA | SIMILKAMEEN RIVER NEAR HEDLEY |
| 02PB006 | 1965 | 2015 | NA | SAINTE-ANNE (RIVIERE) (BRAS DU NORD DE LA) EN AMONT | 05HH001 | 1958 | 2015 | RHOA | SOUTH SASKATCHEWAN RIVER AT ST. LOUIS | 09AE003 | 1956 | 2015 | NA | SWIFT RIVER NEAR SWIFT RIVER |
| 02PJ005 | 1915 | 2015 | RHOA | CHAUDIERE (RIVIERE) AU PONT-ROUTE 218 À SAINT-LAMBERT-DE-LAUZON | 05JM001 | 1915 | 2015 | RHEA | QU'APPELLE RIVER NEAR WELBY | 09AH001 | 1951 | 2015 | NA | YUKON RIVER AT CARMACKS |
| 02QA002 | 1962 | 2015 | NA | RIMOUSKI (RIVIERE) À 3,7 KM EN AMONT DU PONT-ROUTE 132 | 05KC001 | 1955 | 2015 | NA | CARROT RIVER NEAR SMOKY BURN | 09BC001 | 1951 | 2015 | NA | PELLY RIVER AT PELLY CROSSING |
| 02RD002 | 1953 | 2004 | ND | MISTASSIBI (RIVIERE) | 05KH007 | 1965 | 2015 | NA | CARROT RIVER NEAR TURNBERRY | 09BC004 | 1970 | 2015 | NA | PELLY RIVER BELOW VANGORDA CREEK |
| 02RF001 | 1915 | 2015 | NA | ASHUAPMUSHUAN (RIVIERE) À LA TÊTE DE LA CHUTE AUX SAUMONS | 05KJ001 | 1913 | 2015 | RHOA | SASKATCHEWAN RIVER AT THE PAS | 09CD001 | 1956 | 2015 | NA | YUKON RIVER ABOVE WHITE RIVER |
| 02RG005 | 1964 | 2015 | NA | METABETCHOUANE (RIVIERE) EN AMONT DE LA CENTRALE S.R.P.C. | 05LC001 | 1914 | 2015 | NA | RED DEER RIVER NEAR ERWOOD | 09DC002 | 1947 | 1979 | ND | STEWART RIVER AT MAYO |
| 02UC002 | 1965 | 2015 | NA | MOISIE (RIVIERE) À 5,1 KM EN AMONT DU PONT DU Q.N.S.L.R. | 05LH005 | 1923 | 2015 | NA | WATERHEN RIVER NEAR WATERHEN | 09DD003 | 1951 | 2015 | NA | STEWART RIVER AT THE MOUTH |
| 02VC001 | 1956 | 2014 | NA | ROMAINE (RIVIERE) AU PONT DE LA Q.I.T. | 05LM006 | 1967 | 2015 | RHEA | DAUPHIN RIVER NEAR DAUPHIN RIVER | 09EA003 | 1965 | 2015 | NA | KLONDIKE RIVER ABOVE BONANZA CREEK |
| 02WB003 | 1980 | 2015 | NA | NATASHQUAN (RIVIERE) À 0,6 KM EN AVAL DE LA DÉCHARGE DU LAC ALIESTE | 05MD004 | 1944 | 2015 | RHOA | ASSINIBOINE RIVER AT KAMSACK | 09EB001 | 1944 | 2015 | NA | YUKON RIVER AT DAWSON |
| 02XA003 | 1979 | 2015 | NA | LITTLE MECATINA RIVER ABOVE LAC FOURMONT | 05ME006 | 1954 | 2015 | RHOA | ASSINIBOINE RIVER NEAR MINIOTA | 09FB001 | 1965 | 1995 | NA | PORCUPINE RIVER BELOW BELL RIVER |
| 02XA004 | 1979 | 1996 | ND | RIVIERE JOIR NEAR PROVINCIAL BOUNDARY | 05MH005 | 1954 | 2015 | RHOA | ASSINIBOINE RIVER NEAR HOLLAND | 09FC001 | 1976 | 2015 | NA | OLD CROW RIVER NEAR THE MOUTH |
| 02XC001 | 1967 | 2015 | NA | SAINT-PAUL (RIVIERE) À 0,5 KM DU RUISSEAU CHANION | 05NB009 | 1956 | 1995 | RHOD | SOURIS RIVER NEAR ROCHE PERCEE | 09FD001 | 1961 | 1995 | ND | PORCUPINE RIVER AT OLD CROW |
| 02YA002 | 1986 | 2015 | NA | BARTLETTS RIVER NEAR ST. ANTHONY | 05NG001 | 1912 | 2015 | RHOA | SOURIS RIVER AT WAWANESA | 10AA001 | 1960 | 2015 | NA | LIARD RIVER AT UPPER CROSSING |
| 02YK008 | 1985 | 2015 | NA | BOOT BROOK AT TRANS-CANADA HIGHWAY | 05NG021 | 1946 | 2015 | RHOA | SOURIS RIVER AT SOURIS | 10AB001 | 1962 | 2015 | NA | FRANCES RIVER NEAR WATSON LAKE |
| 02YL001 | 1928 | 2015 | NA | UPPER HUMBER RIVER NEAR REIDVILLE | 05OC012 | 1958 | 2015 | RHOA | RED RIVER NEAR STE. AGATHE | 10BB001 | 1960 | 1995 | ND | KECHIKA RIVER AT THE MOUTH |
| 02YO007 | 1984 | 1996 | ND | LEECH BROOK NEAR GRAND FALLS | 05OJ010 | 1960 | 2008 | RHOD | RED RIVER NEAR LOCKPORT | 10BB002 | 1967 | 1994 | ND | KECHIKA RIVER ABOVE BOYA CREEK |
| 02YO012 | 1989 | 2015 | NA | SOUTHWEST BROOK AT LEWISPORTE | 06AD001 | 1933 | 2015 | NA | BEAVER RIVER NEAR DORINTOSH | 10BE001 | 1944 | 2015 | NA | LIARD RIVER AT LOWER CROSSING |
| 02YQ004 | 1983 | 1998 | ND | NORTHWEST GANDER RIVER NEAR GANDER LAKE | 06AD006 | 1955 | 2015 | NA | BEAVER RIVER AT COLD LAKE RESERVE | 10BE005 | 1960 | 2015 | ND | LIARD RIVER ABOVE BEAVER RIVER |
| 02ZD002 | 1969 | 2015 | NA | GREY RIVER NEAR GREY RIVER | 06AG001 | 1971 | 2015 | NA | BEAVER RIVER BELOW WATERHEN RIVER | 10BE006 | 1969 | 1995 | ND | LIARD RIVER ABOVE KECHIKA RIVER |
| 03BF001 | 1975 | 2015 | NA | PONTAX (RIVIERE) À 60,4 KM DE L'EMBOUCHURE | 06BC001 | 1970 | 1995 | ND | MUDJATIK RIVER NEAR FORCIER LAKE | 10CC002 | 1978 | 2004 | ND | FORT NELSON RIVER ABOVE MUSKWA RIVER |
| 03CB001 | 1959 | 1980 | ND | EASTMAIN (RIVIERE) EN AVAL DE LA RIVIERE A L'EAU CLAIRE | 06BD001 | 1966 | 2015 | NA | HAULTAIN RIVER ABOVE NORBERT RIVER | 10CD001 | 1944 | 2015 | NA | MUSKWA RIVER NEAR FORT NELSON |
| 03CB004 | 1979 | 2004 | ND | EASTMAIN (RIVIERE) A LA TETE DE LA GORGE PROSPER | 06DA004 | 1966 | 2015 | NA | GEIKIE RIVER BELOW WHEELER RIVER | 10EA003 | 1960 | 2015 | NA | FLAT RIVER NEAR THE MOUTH |
| 03CC001 | 1958 | 1980 | ND | EASTMAIN (RIVIERE) A LA TETE DE LA GORGE DE BASILE | 06GD001 | 1955 | 2015 | NA | SEAL RIVER BELOW GREAT ISLAND | 10EB001 | 1960 | 2015 | NA | SOUTH NAHANNI RIVER ABOVE VIRGINIA FALLS |
| 03DD002 | 1960 | 1993 | ND | DE PONTOIS (RIVIERE) EN AMONT DE LA RIVIERE SAKAMI | 06JC002 | 1965 | 2015 | NA | THELON RIVER ABOVE BEVERLY LAKE | 10EC001 | 1959 | 1996 | ND | SOUTH NAHANNI RIVER ABOVE CLAUSEN CREEK |
| 03ED001 | 1961 | 2015 | NA | BALEINE (GRANDE RIVIERE DE LA) EN AMONT DE LA RIVIERE DENYS-1 | 06LC001 | 1960 | 2015 | NA | KAZAN RIVER ABOVE KAZAN FALLS | 10ED001 | 1942 | 2015 | NA | LIARD RIVER AT FORT LIARD |
| 03HA001 | 1954 | 1963 | ND | ARNAUD (PAYNE)(RIVIERE) EN AMONT DE LA RIVIERE HAMELIN-1 | 06MB001 | 1969 | 1996 | ND | QUOICH RIVER ABOVE ST. CLAIR FALLS | 10ED002 | 1972 | 2015 | NA | LIARD RIVER NEAR THE MOUTH |
| 03JB001 | 1955 | 1988 | ND | FEUILLES (RIVIERE AUX) EN AVAL DE LA RIVIERE PELADEAU | 07AE001 | 1960 | 2015 | NA | ATHABASCA RIVER NEAR WINDFALL | 10GB006 | 1974 | 2015 | NA | WILLOWLAKE RIVER ABOVE METAHDALI CREEK |
| 03KC004 | 1965 | 2015 | NA | MELEZES (RIVIERE AUX) À 7,6 KM EN AMONT DE LA CONFLUENCE AVEC LA KOKSOAK | 07BC002 | 1957 | 2015 | NA | PEMBINA RIVER AT JARVIE | 10GC001 | 1938 | 2015 | RHEA | MACKENZIE RIVER AT FORT SIMPSON |
| 03MB002 | 1956 | 2015 | NA | BALEINE (RIVIERE A LA) À 40,2 KM DE L'EMBOUCHURE | 07BE001 | 1913 | 2015 | NA | ATHABASCA RIVER AT ATHABASCA | 10HB005 | 1975 | 2015 | NA | REDSTONE RIVER 63 KM ABOVE THE MOUTH |
| 03MC001 | 1972 | 1993 | ND | TUNULIC (RIVIERE) PRES DE L'EMBOUCHURE | 07CD001 | 1930 | 2015 | NA | CLEARWATER RIVER AT DRAPER | 10KA001 | 1943 | 2015 | RHEA | MACKENZIE RIVER AT NORMAN WELLS |
| 03MD001 | 1975 | 2015 | NA | GEORGE (RIVIERE) À LA SORTIE DU LAC DE LA HUTTE SAUVAGE | 07DA001 | 1957 | 2015 | NA | ATHABASCA RIVER BELOW FORT MCMURRAY | 10LA002 | 1968 | 2015 | NA | ARCTIC RED RIVER NEAR THE MOUTH |
| 03NF001 | 1978 | 2015 | NA | UGJOKTOK RIVER BELOW HARP LAKE | 07EA005 | 1978 | 2015 | NA | FINLAY RIVER ABOVE AKIE RIVER | 10LC002 | 1972 | 2015 | RHOA | MACKENZIE RIVER (EAST CHANNEL) AT INUVIK |
| 03NG001 | 1977 | 1996 | ND | KANAIRIKTOK RIVER BELOW SNEGAMOOK LAKE | 07EC002 | 1975 | 2015 | NA | OMINECA RIVER ABOVE OSILINKA RIVER | 10LC014 | 1985 | 2015 | RHOA | MACKENZIE RIVER AT ARCTIC RED RIVER |
| 03PB002 | 1977 | 2015 | NA | NASKAUPI RIVER BELOW NASKAUPI LAKE | 07FB001 | 1961 | 2015 | NA | PINE RIVER AT EAST PINE | 10MA001 | 1961 | 2015 | NA | PEEL RIVER ABOVE CANYON CREEK |
| 03QC001 | 1966 | 2015 | NA | EAGLE RIVER ABOVE FALLS | 07FC001 | 1917 | 2015 | NA | BEATTON RIVER NEAR FORT ST. JOHN | 10MC002 | 1969 | 2015 | NA | PEEL RIVER ABOVE FORT MCPHERSON |
| 03QC002 | 1978 | 2015 | NA | ALEXIS RIVER NEAR PORT HOPE SIMPSON | 07GE001 | 1917 | 2015 | NA | WAPITI RIVER NEAR GRANDE PRAIRIE | 10NC001 | 1969 | 2015 | NA | ANDERSON RIVER BELOW CARNWATH RIVER |
| 04AB001 | 1972 | 2015 | NA | HAYES RIVER BELOW GODS RIVER | 07GH002 | 1959 | 2015 | NA | LITTLE SMOKY RIVER NEAR GUY | 10QC001 | 1976 | 2015 | NA | BURNSIDE RIVER NEAR THE MOUTH |
| 04AD002 | 1967 | 2015 | NA | GODS RIVER NEAR SHAMATTAWA | 07GJ001 | 1915 | 2015 | NA | SMOKY RIVER AT WATINO | 10QD001 | 1969 | 2015 | NA | ELLICE RIVER NEAR THE MOUTH |
| 04CC001 | 1968 | 1995 | ND | SEVERN RIVER AT LIMESTONE RAPIDS | 07HA001 | 1915 | 2015 | RHEA | PEACE RIVER AT PEACE RIVER | 10RA001 | 1977 | 2015 | NA | BACK RIVER BELOW BEECHY LAKE |
| 04DC001 | 1965 | 2015 | NA | WINISK RIVER BELOW ASHEWEIG RIVER TRIBUTARY | 07HA005 | 1967 | 2015 | NA | WHITEMUD RIVER NEAR DIXONVILLE | 10RA002 | 1977 | 2015 | NA | BAILLIE RIVER NEAR THE MOUTH |
| 04EA001 | 1967 | 2015 | NA | EKWAN RIVER BELOW NORTH WASHAGAMI RIVER | 07HC001 | 1961 | 2015 | NA | NOTIKEWIN RIVER AT MANNING | 10RC001 | 1960 | 2015 | NA | BACK RIVER ABOVE HERMANN RIVER |
| 04FC001 | 1968 | 2015 | NA | ATTAWAPISKAT RIVER BELOW MUKETEI RIVER | 07JD002 | 1970 | 2015 | NA | WABASCA RIVER AT HIGHWAY NO. 88 | 10SB001 | 1971 | 1994 | ND | HAYES RIVER ABOVE CHANTREY INLET |
| 04GD001 | 1966 | 2015 | RHOA | ALBANY RIVER ABOVE NOTTIK ISLAND | 07KC001 | 1959 | 2015 | RHEA | PEACE RIVER AT PEACE POINT (ALBERTA) | 11AA005 | 1909 | 2015 | RHEA | MILK RIVER AT MILK RIVER |