# Peer review of "A Canadian River Ice Database from National Hydrometric Program Archives"

_Earth System Science Data, 2020_

## Referee Comment (RC1) · Benoit Turcotte (Referee) · 19 Apr 2020

Dear Authors,

I am pleased to provide a review for this paper. It represents a tremendous amount of work and a significant publication that will generate a positive impact on river ice research in years to come. I will definitely use the CRID and it will soon become a widely understood acronym within the river ice community and Canada and abroad. I know how it feels to analyze hundreds and hundreds of data sets, and then having to do it again in the most consistent way possible because of the need to to add another winter variable.

General comments:

[Figure]

- The tone of the introduction could be slightly adapted (see specific comments). It is true that this type of publication and database has not been seen in the past, but I believe that the absence of a CRID before now never prevented meaningful research to be completed and published. The hydrometric data has always been accessible and it was analyzed as needed. A research paper about ice processes at a specific location is valuable and should not be overlooked because it only includes data from a single or a handful of sites.

- The authors could state more formally (in the Data Disclaimer, but also elsewhere in the paper) that even a data rating of 0 may not replace the Engineer's professional responsibility for the conception of flood maps and for the design of hydraulic structures.

Specific comments: The sum of the experience of all authors is spectacular, and these comments will hopefully be perceived as constructive. Most of them are suggestions. There are lots of comments, but I am really taking this at heart and hope that this publication can be as perfect as possible.

Lines 17-18: This is a typical expression used on the Canadian West Coast, in Southern Ontario or in Eastern and central United States. River ice is not only common in cold regions, it is a part of the annual cycle, like open water conditions. I suggest rewording this.

Line 18-19: Not sure why this sentence is here. There has been papers focusing on many sites and many rivers. In turn, there is a reason why specific reports try to address local issues. In both cases, the Canadian data base would be useful

Line 36: Why not saying : River ice processes are an intrinsic component of cold climate watersheds.

Lines 37-41: The authors could refer to CRIPE at this point in the introduction. This Canadian research group on river ice has been quite active and productive since the 1980.

Line 42: Following the general comment #1, I am not sure why this sentence starts with "However"

Lines 43-44: This is not necessarily true. Researchers have been extracting the data that they needed, most of the time. It has just not been done in a consistent way.

Line 50: "calculating" could be "estimating". Using "calculation" may insinuate that the result is exact, which is not the case.

Line 50-52: This is the main point of the paper.

Lines 69-77: Note that these examples are all from the Mackenzie basin, and then, the following paragraph is about outside Canada. Should there be a short mention of river ice studies in other watersheds in Canada before initiating the following paragraph?

Lines 90-93: This comes back to Canada. Scandinavia is not mentioned in this paragraph. They must have done similar work, and if not, it could be mentioned.

Lines 99-101: Indeed, no one has ever done an extraction of all river ice variables on so many Canadian rivers. This should not be expressed as a weakness from the literature, but as a strength of this research to support other research and development. This paper is strong enough to avoid falling on the classic message about the need to fill obvious gaps in the literature.

Line 168: Can you please double-check that Groudin is not Grondin (a more common name)? Also in Table 1. You may very well be correct.

Line 194: Is "potential" the right word here? My understanding of potential is what can be reached or achieved at a site or station, as opposed to the fine-scale maximum at a station for any given year.

Line 197: Should "daily" be "daily-averaged"?

Line 199: Should "depends" be "depending"?

[Figure]

Figure 3: You could clarify this figure by adding the duration of the ice season. I am not sure that the title of the X axis is accurate. This cannot be a complete year, at least not if the scale is constant. Last B date is quite low compared with HM. Is this a typical behavior? I like that HO is significantly lower that HM, but again, is this typical? It just seems that so much water has been flowing during breakup and that the freshet is almost over by then. I understand that this may be representative of a specific river, but is this largely applicable / representative of Canadian River?

Figure 3: The peak to and from HM is intriguing to me. It is a relatively gradual rise, which does not suggest the formation of an ice jam. Then, the water level drop does suggest the gradual thermal melting of an ice jam. Also, in my mind, Last B date should be at higher level than HB, but I may be wrong.

HLQ1:Not sure if this is well positioned in Fig. 3. It seems that after freeze-up, thermal thickening or thermal erosion should follow. Therefore, I do not see why this first minimum Q would occur during the subsequent rise in water level. I may be wrong and you may have seen this at some stations.

Table 2: First B date (and last B date): Has this been re-analyzed or indicated B dates were just adopted as they appeared in reports? My understanding is that B dates are often off by a few days and this can be checked with some temperature and hydrological indicators. It can have a significant impact when preparing flood maps that distinguish different flooding processes.

HF: This is quite obvious when the ice cover forms by frontal progression, but the gradual formation of border ice followed by ice congestion in a relatively narrow open water channel may not generate a clear signal. That being said, there would most probably always be a "maximum freeze-up level", and this may be a more appropriate name for this parameter. (I am unsure how you would differentiate that from a small runoff event taking place during freeze-up and generating or not, a freeze-up jam.). I appreciate the explanation provided at lines 362-368.

[Figure]

HF2: Could change the name of this variable to "water level at second freeze-up"

Line 258: Drifting ice is part of the flow, it is not stagnant ice, and it should not generate backwater if the surface concentration remains low. Same comment for flowing ice chunks.

Line 284: I am unsure why point C is not at the first spike that seem to be sharp enough to represent a local ice movement, possibly a downstream partial breakup that would reduce backwater at the station.

Lines 290-291: I believe that hydrological simulation, comparison with other stations, or judgment can still provide some kind or error margin (it can hardly be more than one order of magnitude, at least).

Lines 307-311: Not sure if this paragraph invites CRID users to report on possible errors that could justify specific re-analyses. I believe that it should be the case, but it depends on how ECCC will want to maintain and update the CRID.

Line 327: The authors could confirm if this first B date on Oct 10 was the result of a rise in water level that trigger the decision to initiate B condition. More generally, the authors could confirm the information (cameras?) or signs (rise in water level after X degree-days of freezing) that are usually considered to initiate B conditions.

Figure 5: Adding the water level signal to this figure would be of interest, but it does represent some work.

Lines 351-352: The authors could mention something about peak factors (instantaneous divided by daily-averaged) here. For freeze-up, peak factors can be in the order of 1.1 or 2.0, depending on freeze-up dynamics... This would just be a reminder that using a 1 for design may be unsafe.

Line 367: The authors could mention that this can take place over a distance of several hundred km upstream.

[Figure]

Line 371: "...snow and..." First time I see this expression. In some regions, there is snow, but no ice cover because there is too much heat (downstream of lakes or reservoirs, or maybe downstream of cities and industries). Still, the word "snow" here may create confusion since this paper is about the ice cover period.

Lines 377-379: It could be mentioned that this is common and mostly caused by the thicker ice cover at the end of winter that generates a higher water level despite this being the actual winter min Q.

Line 383: "analysis" is probable "analyses" (plural)

Line 396: "risk" and "threats": The risk cannot be a threat. Consider rephrasing this considering that the risk is a combination of consequence and probability (or possibility) of a hazard and that a threat is in this case a hazard.

Lines 400-401: There are also records from nearby hydrometric stations.

Line 407: Not sure if a sudden drop in water level can be considered as a "spike". Also, depending on where the station is located and about the intensity of the winter runoff events, the water level signal can be a drop (local breakup) , a gradual rise (ice cover is lifted), a sudden rise (ice jam formation), or a combination of the aboves.

Line 411: In areas where multiple mid-winter breakup events occur, they can be hard to distinguish from freeze-up chaos. First question: Does a mid-winter runoff event only qualifies after a complete ice cover has formed? Second question: why not using the highest mid-winter peak instead of the first one? Third question: How would you consider a massive breakup event at the end of February like it happened in 1981 in southern and central Quebec? Would that be a mid-winter breakup followed by no more winter, or would that be the spring breakup event? I am curious, but understand that we may not have to start a conversation about this.

Lines 414-415: I am not too familiar with WSC's practice, but I would be very careful to remove a B in the middle of winter following a mid-winter breakup event. This may

occur in NE, NB, southern QC and ON as well as in West-southern BC, but in most of Canada, after a complete mid-winter breakup, the presence of shear walls would prevent the removal of the B until the flow has receded significantly and this is when a cold spell may have already created border ice.

Line 431: "mark": I would say "may mark" as this is not the case for all types of rivers.

Line 435: Depends where: In some cases, a mid-winter breakup event is followed by a dramatically cold period during which frazil generation is significant. The result may be a very thick ice accumulations, with inflated ice jams and new anchor ice cycles.

Line 436: Of course, daily-averaged levels may appear smooth enough. At specific locations, the water level could remain high or even increase even though the discharge drops. This would be caused by progressive frazil accumulation produced in a newly open (steep) reach exposed to cold air. Hydrometric stations are usually not located in reaches affected by this type of process. I am just providing this information in case it would seem appropriate to adapt the text (and this applies to many other comments).

Lines 450-452: A hanging dam can form several km downstream of an open reach. It all depends on the river gradient and profile. In the case of anchor ice, it can hardly remain in place for several months. It will either contribute to the formation of a complete surface ice cover, or will melt away during mild spells and come back during cold spells. I suggest that this creeping signal is mostly associated with frazil accumulation.

Lines 452-453: But wouldn't it still deplete during the winter time? I see that you have a reference at the end of the sentence but does that reference suggest that?

Figures 8 and 9: A superposed air temperature graph would be of interest.

Line 471: Please update Figure #

Line 485: "Impure ice": Is this common? Should you explain what this means in brackets? Should you also add this to the previous sentence that refers to snow load, for consistency?

Lines 504-506: You may suggest that readers could take the measured thickness and associated date, evaluate the corresponding cumulated degree-days of freezing (or a cumulated sophisticated heat budget), and create a relationship between both parameters. Step 2 would simply be to apply this relationship to the maximum degree-days of freezing of each winter to obtain an estimate of the maximum ice thickness (if no mid-winter breakup occurred between ice thickness measurement and actual max freezing degree-days).

Lines 517-518: Actually, the station may start "feeling" some stage instabilities that come from upstream (these would actually be discharge instabilities induced by upstream ice movement), and it would still mean that breakup has initiated. How do we know that this is taking place downstream, especially when looking at daily-average stage data?

Line 518: Same comment as before: a reduction in roughness would generate a sudden drop of the instantaneous stage signal. In turn, a jave would be a spike and a sudden raise would be the formation of an ice jam downstream.

Line 529: I am not sure that there is a need to state "quickly" here. First, it applies to both time and distance traveled. Second, quickly is relative and I have seen large ice slabs (especially those that were part of a hanging dam or a snowmobile crossing) remaining fairly large several days or km after breakup.

Line 532: Should there be an example of a case study reporting X meters above the rating curve? This would illustrate the meaning of "far exceed"

Lines 534-535: This is not exact: They can also cause a measurable stage (actual discharge) depressions for several hours before reaching an equilibrium. The jave is much more sharp, especially in steep channels and when the released jam was not too far upstream from the station.

Lines 535: It should be stated that 1. Javes can only be adequately documented using

instantaneous data. 2. Javes have probably been removed from discharge records (at least in Quebec) as they were considered to be ice jams that had nothing to do with a discharge signal. It is also possible that javes and ice jams have been removed from some records because they were perceived as instrument pathologies. If there is enough evidence of this practice in some offices, the authors should mention it in the discussion.

Line 541: Could be completed by "... where the stage gradually returns to the stage-discharge relationship as the discharge slowly increases"

Line 550: Should the authors state that the last B date could likely be off by a few days? It is not to criticize the work done by different offices, but to warn users about this possible limitation. The last B date is specially difficult to confirm during thermal breakup years or when post-break ice runs from far upstream still occur after a complete local wash.

Figure 12: Second image is very dark. Is there a way to tweak this?

Line 575: Should the authors mention that Ho may actually occur in mid-summer (e.g., Saguenay event in Quebec, 1996) or during the fall, and therefore may not be associated with the spring freshet, especially in Eastern Canada?

Line 577-578: Just to complete the idea, i would suggest: "...for a large ratio of hydrometric stations in Canada, and most probably for an equal ratio of unmonitored sites."

Line 584: "five" should probably be "six"

Line 605: Should the author specific what defines an error or what it the calculation behind this %?

Lines 606-607: It is unclear to me if indicated B dates are considered true and other parameters are corrected consequently, or the opposite.

Lines 613-614: As asked earlier, would the authors also commit to present updated

versions of the CRID with corrections?

Lines 623 vs. line 634: If I had to choose, I would say that ice processes are site specific.

Line 628-629: (e.g. promoting a thicker ice cover in the deck shadow and promoting ice jamming against abutment or pillars)

Figure 13: The legend in this graph could include variable acronyms for clarity. Also, it would have been useful to separate the two populations with different icons / colors. The only obvious difference is the two populations is blue circles.

Table 4: There may not be enough space, but the authors could consider adding a column with the variable acronym.

Line 694: "Very often" Do we have an updated number about that? If not, I hope that the CRID will be used by researchers to update the one third presented by Beltaos years ago.

Line 694-695: I am not sure that I agree with this interpretation. It can be said that ice jams produce higher water levels at similar high flows (quite logical), and it can be said that at some sites, the main flooding process is caused by ice processes. In turn, the highest discharge in rivers most often occur in the absence of ice. There should be a more efficient way to express this.

Line 696: "eg." should be "e.g.". I take note that FloodNET is only one example. Other groups have completely ignored river ice processes in their flood research.

Line 700: "could likely" should be "should, when applicable,"

Line 701: For sites that are not included in the CRID and where winter water level information is available, the CRID can represent a template to extract pertinent information for various purposes, including flood mapping and hydraulic structure design.

Line 742: A last sentence could be: "Maintaining funding and constantly improving

hydrological estimation and measurements approaches is needed to maintain an adequate level of knowledge and to update the CRID in the future."

Lines 1042-1044: I do not see this paper referred to in the paper and it should removed from the reference.

---

## Referee Comment (RC2) · Anonymous Referee #2 · 23 Apr 2020

Title: A Canadian River Ice Database from National Hydrometric Program Archives

Author(s): Laurent de Rham, Yonas Dibike, Spyros Beltaos, Daniel Peters, Barrie Bonsal, Terry Prowse

MS No.: essd-2020-29

General Comments:

The manuscript introduces the newly developed Canadian River Ice Database (CRID). Such a database is very welcomed in the river ice science and practitioner community and will promote studies to address a variety of research questions and practical issues. It is tremendous efforts to go through the large amount of historical data and collect the variables related to specific key ice events. Several of these variables can be very challenging to identify and require extensive expertise in river ice engineering, which is offered by the author team. The team's experience and expertise are also reflected in the selection of the variables, detailed description of their physical importance, quality control of the ice data, and uncertainty assessment. In this regard, the manuscript provides an important reference document for the use of the CRID.  I will definitely be using the database and would like to see it being updated regularly as new information becomes available.

Specific comments:

Line 87: select to selected

Line 115-126: It seems that with minimum 20-year record, no minimum drainage area and including both north of 0deg isotherm and southern temperate zone would result in much more than 196 stations. Am I missing any additional selection criteria used here?

Line 135: foci to forcing

Line 139: listing to list

Line 191: There are actually more than 15 variables as several of the ones listed in Table 2 include both water level and discharge and they probably should be counted as 2 variables.

Figure 3: I am not sure if this figure is based on actual gauge record or purely conceptual. It may worth to show a water level hydrograph where the key ice events are less obvious (less "spiky") and explain how the different variables are identified.

Table 2: does the wording "data accuracy" best represent what this indicator really means? It may lead reader/user to think the published data is accurate while it is less likely in case of ice affected discharge data.

Section 3.3 What are the methods used to compute discharges under ice conditions? Can the authors briefly describe some common ones? This is important information for users of the published discharge data. Additionally, my understanding is that different methods and techniques have been used when deciding when to start and end the B symbol. Maybe the authors can provide some information on this as well?

Line 278: repetitive quotation marks

Page 12: Section goes from 3.3 to 3.4.1, missing 3.4

Line 345-348: It may not be accurate to say the initial ice cover progression past a gauge is always a spike in the water level chart. In many cases, the "stage up" caused by an ice cover approaching from downstream and passing a gauge is a gradual water level increase. How is HF decided in a case like this?

Line 483: maybe add "approximately" before 0.92 as ice density can be affected by many factors.

Line 501 Fig. 10 should be Fig. 11

Line 517-518: this statement about the spike on the water level hydrograph indicating the onset of breakup seems to be conflicting with line 539-541. In the case of thermal breakup, how is HB determined?

Line 529-531: ice jams can form at morphologically conducive locations even without intact ice cover stopping the ice run.

Line 534-535: Jams formed upstream of a gauge may also choke the flow. It also depends on its vicinity to the gauge.

Line 545 chuck -> chunk

Line 553-556: I wouldn't say the last B date is always used as a surrogate/index, and less accurate than the CRID data to analyze spring breakup timing. They just represent different stage of the breakup.

Line 573-575: how can one calculate the water level using rating curve when instrumentation is damaged or not functioning?

Line 603-607 unclear to me how the percentage error are calculated.

Line 635 Fig 12 should be Fig. 13

---

## Referee Comment (RC3) · Zoe Li (Referee) · 30 Apr 2020

Comments to the Editor:

The authors developed a Canadian River Ice Database using the Canadian National Hydrometric Program hydrometric records. River ice related events, especially ice jam flooding, are of great importance to the watershed management in many cold regions around the world, including Canada. This database provides a significant amount of valuable data to support river ice research and applications. I can definitely see myself and my colleagues using this database. This paper is well organized and well written. I only have some minor concerns as indicated in the comments to the authors. I suggest a minor revision.

[Figure]

Detailed Comments to the Authors:

- Line 24: "73,000 variables" should be changed to "73,000 records".

- Line 28: "a time series of up to 15 variables" should be changed to "time series of up to 15 variables".

- Lines 119-126: It is not clear how the 196 sites in this database were selected. Does it include any of the additional 60 southern sites? Or is it the same 196 gauging stations as in the NHP archives?

- Line 135: Typo: "hydro-ecological foci".

- Figure 2: Consider removing the border lines and using a different color for stations not in operation. - Table 1: Add bottom border.

- Figure 3: Add a legend for the grey line to show it is the water level during mid-winter breakup.

- Line 265: It's not clear which 12 discharge time series the authors meant.

- Line 315: The subtitle of section 3.4 is missing.

- Line 325 & Figure 5: Consider defining the colors in the MODIS images for readers who are not familiar with satellite images.

- Line 333: An extra space in "Sect. 3.4.4 )".

- Line 365: "parameterizes" should be changed to "parameterize".

- Line 466: An extra space in "level ."

- Line 496: No need to provide the abbreviation S.T.B. if it is used only once in the manuscript.

- Line 512: An extra space in "(84 days after January 1) ."

- Line 618: An extra space in "about 1 hour ."

- Table 2: Change "2000-01" to "2000-2001".

- Tables 2 and 3: The column heads need to be re-formatted.

- Lines 365-368: It is not quite clear why the length of water level data was determined to be 30 days.

- Line 412: What about HMWB? How was it determined when there are no continuous water level records?

- In Section 3.4, the variables were classified into 7 groups (7 subsections). Reasoning for the classification should be provided and reflected in the subtitles.

- A brief data management plan, particularly the current database maintenance and update plan, should be provided.

- There are some minor formatting errors in the references section. For example, the format of doi is not consistent. All references should be provided in the same format.

---

## Author Comment (AC1) · 22 Jun 2020

Author Responses to Referee's Comments

The authors would like to thank all three reviewers for the time and effort reviewing this manuscript (MS). The reviews are favorable with indication that this database will be widely used. We appreciate the numerous suggestions, to which the vast majority have been incorporated into the revised MS. This document compiles: (1) comments from referees, (2) author's response and (3) author's changes in manuscript that include the page, line and a statement on the revision.  First, Reviewer 1: Benoit Turcotte (RC1) comments are followed by author response (AR1) and applicable changes (Action1). Then, Reviewer 2: Anonymous Referee (RC2) comments are followed by author response (AR2) and applicable changes (Action2).  Finally, Reviewer 3: Zoe Li (RC3) comments are followed by author response (AR3) and applicable changes (Action3).  In a few cases, where no changes are made, we state 'No Action' followed by appropriate reviewer number.
A revised version of the MS using Track Changes is included. The page and line number refer to those in the Track Changes version when 'All Markup' is selected.
After addressing all the reviews comments, we did a final read through and make a few minor editorial changes to the text, figures and tables to improve the readability of the MS.

We have added the following statement in Acknowledgments: Page 45, Line 900-901: "Thank you very much to Dr. Benoit Turcotte, Dr. Zoe Li and  Anonymous Referee for the detailed reviews that improved this manuscript.
I am pleased to provide a review for this paper. It represents a tremendous amount of work and a significant publication that will generate a positive impact on river ice research in years to come. I will definitely use the CRID and it will soon become a widely understood acronym within the river ice community and Canada and abroad. I know how it feels to analyze hundreds and hundreds of data sets, and then having to do it again in the most consistent way possible because of the need to  add another winter variable.

AR1: We appreciate Dr. Turcotte's comprehensive review of the manuscript. The feedback is very constructive and it is encouraging to hear a vision for the CRID from a prospective user.

No Action1

General comments:

RC1: The tone of the introduction could be slightly adapted (see specific comments). It is true that this type of publication and database has not been seen in the past, but I believe that the absence of a CRID before now never prevented meaningful research to be completed and published. The hydrometric data has always been accessible and it was analyzed as needed. A research paper about ice processes at a specific location is valuable and should not be overlooked because it only includes data from a single or a handful of sites.

AR1: The authors acknowledge and appreciate the excellent research over the years on river ice processes at specific locations and have no intention of diminishing the value of those works. We just want to indicate that there is no previous pan-Canadian river ice study because of the lack of a national database. Therefore, the objective of this database/paper is to compile and report on Canada wide river ice information from NHP archives. This follows on Beltaos and Prowse (2009) recommendations described on page 3, line 108-14. We have now included additional introductory material to highlight the importance of other river ice science and activities.

Action1: Page 1, line 20/21 – added statement: "single locations or regional assessments, are season-specific and use readily available data"

Action1: Page 2, Line 47-52 – added statement: "While there are growing number of publications on river ice processes focusing on specific locations or river reaches and looking at a specific part of the ice period, such as the spring break-up, there are only few large-scale (countrywide) studies on the complete river ice season because of the absence of a comprehensive and multi-site river ice database. It is not commonly known by the wider hydrology research community that a valuable source on river ice information can be extracted from the archives of hydrometric networks."

Action 1, Page 3, line 87-89 – added statement: Other well studied Canadian locations include, to mention but a few, the Hay River (De Coste et al., 2017); Red River (Wazney and Clark, 2015) and Chaudiere River (De Munck et al., 2016).

Action1: Page 3, line 102/103 – added statement: "A compilation and analysis of Norwegian rivers ice  was described by Gebre and Alfredsen (2011)."

RC1: The authors could state more formally (in the Data Disclaimer, but also elsewhere in the paper) that even a data rating of 0 may not replace the Engineer's professional responsibility for the conception of flood maps and for the design of hydraulic structures.

AR1: The Data Disclaimer used is based on generic ECCC standard. We thank the reviewer for his recommendation and have incorporated this in the text.

Action 1: Page 38, Line 717-718 – add sentence: "The data quality ratings should not replace the professional responsibility of engineers and geoscientists for the conception of flood maps and for the design of hydraulic structures"

RC1: Specific comments: The sum of the experience of all authors is spectacular, and these comments will hopefully be perceived as constructive. Most of them are suggestions. There are lots of comments, but I am really taking this at heart and hope that this publication can be as perfect as possible.

AR1: Thank-you. This review is very constructive and improves the manuscript

RC1: Lines 17-18: This is a typical expression used on the Canadian West Coast, in Southern Ontario or in Eastern and central United States. River ice is not only common in cold regions, it is a part of the annual cycle, like open water conditions. I suggest rewording this.

AR1: Agree with the reviewer, remove word "common" and revise to:

Action1: Page 1, Line 17-18: reword  sentence: "River ice, like open water conditions, is an integral component of the cold climate hydrological cycle. The annual succession… "

RC1: Line 18-19: Not sure why this sentence is here. There has been papers focusing on many sites and many rivers. In turn, there is a reason why specific reports try to address local issues. In both cases, the Canadian data base would be useful

AR1: With this statement, we are stating the fact that, other than few studies that assessed B dates, river ice studies based on many sites and many rivers are not common in the river ice literature. We wanted to indicate that such studies were not common since there was no Canada wide river ice data base, and we are now trying to fill that gap by compiling the CRID.

Action1: Page 1, Line 20-21: revise to say "Reports and associated data on river ice occurrence are often limited to single locations or regional assessments, are season-specific and use readily available data."

RC1: Line 36: Why not saying: River ice processes are an intrinsic component of cold climate watersheds.

AR1: Agree.

Action1: Page 2, Line 37: revise sentence to "River ice is an intrinsic component of cold climate watersheds"

RC1: Lines 37-41: The authors could refer to CRIPE at this point in the introduction. This Canadian research group on river ice has been quite active and productive since the1980.

AR1: Agree. We note that CRIPE active since 1970s' and revise text as:

Action1: Page 2, Line 42-45: Revised to say "The Committee on River Ice Processes and the Environment (CRIPE; http://www.cripe.ca/) has been quite active and productive since the 1970s (Beltaos, 2012a) as the study of river-ice processes and hydraulics emerged as an important research area (Hicks, 2008), while the past decade includes a renewed focus on its ecological aspects (e.g., Peters et al., 2016; Lindenschmidt et al., 2018).

Add to reference:

Beltaos, S. Canadian Geophysical Union Hydrology Section Committee on River Ice Processes and the Environment: Brief History. Journal of Cold Regions Engineering, 26(3), 71–78, 2012a

Action1: additional Beltaos 2012 requires 'b' to be added to this ref and appropriate ref in text at Figure 4
Beltaos, S.: Mackenzie Delta flow during spring breakup: Uncertainties and potential improvements. Canadian Journal of Civil Engineering, 39, 5, 579-588, https://doi.org/10.1139/l2012-033, 2012b

RC1: Line 42: Following the general comment #1, I am not sure why this sentence starts with "However"

AR1: Agree.

Action1: Page 2, Line 47-50: Revised to start with "While there are growing number of publications on river ice processes focusing on specific locations or river reaches and looking at a specific part of the ice period, such as the spring break-up, there are only few large-scale (countrywide) studies on the complete river ice season because of the absence of a comprehensive and multi-site river ice database."

RC1: Lines 43-44: This is not necessarily true. Researchers have been extractting the data that they needed, most of the time. It has just not been done in a consistent way.

AR1: Revise to incorporate 'wider hydrology research community'

Action 1: Page 2, Line 51-52:  Revise to say: "It is not commonly known by the wider hydrology research community that a valuable source on river ice information can be extracted from the archives of hydrometric networks"

RC1: Line 50: "calculating" could be "estimating". Using "calculation" may insinuate that the result is exact, which is not the case.

AR1: Agree.

Action1: Page 2, Line 58: revise  to say: "…for estimating channel discharge."
Action 1: Page 2, line 59: revise to say: "… when producing discharge estimates."

RC1: Line 50-52: This is the main point of the paper.

AR1: Agree.

No Action1

RC1: Lines 69-77: Note that these examples are all from the Mackenzie basin, and then, the following paragraph is about outside Canada. Should there be a short mention of river ice studies in other watersheds in Canada before initiating the following paragraph?

AR1: The examples provided here focus towards studies that specifically used CRID data.  We have now added a few more examples of river ice studies in other watersheds in Canada:

Action1: Page 3, line 87-89: Added sentence: "Other well studied Canadian locations include, to mention but a few, the Hay River (De Coste et al., 2017); Red River (Wazney and Clark, 2015) and Chaudiere River (De Munck et al., 2016)."

And added following to reference list:
De Coste, M., She, Y., Blackburn, J. : Incorporating the effects of upstream ice jam releases in the prediction of flood levels in the Hay River delta, Canada, Canadian Journal of Civil Engineering, 44(8) 643-651, https://doi.org/10.1139/cjce-2017-0123, 2017

Wazney, L., and Clark, S.P.:  The 2009 flood event in the Red River Basin: Causes, assessment and damages, Canadian Water Resources Journal, 41(1-2), 56-64, https://doi.org/10.1080/07011784.2015.1009949, 2015

De Munck, S., Gauthier, Y., Bernier, M., Chokmani, K., Légaré, S.: River predisposition to ice jams: a simplified geospatial model. Natural Hazards and Earth System Sciences Discussions, 17(7), 1033-1047, https://doi.org/10.5194/nhess-17-1033-2017, 2016

RC1: Lines 90-93: This comes back to Canada. Scandinavia is not mentioned in this paragraph. They must have done similar work, and if not, it could be mentioned.

AR1: Agreed. We have included reference to Scandinavia study:

Action1: Page 3, line 102-103: Added sentence: "A compilation and analysis of Norwegian rivers ice  was described by Gebre and Alfredsen (2011)"

And added following to reference list:
Gebre, S.B, and Alfredson, K.T.: Investigation of river ice regimes in some Norwegian water courses, in Proceeding of the 15$^{th}$ Workshop on the Hydraulics of Ice Covered Rivers, Winnipeg, Manitoba, Canada. http://www.cripe.ca/publications/proceedings/16, 2011

RC1: Lines 99-101: Indeed, no one has ever done an extraction of all river ice variables on so many Canadian rivers. This should not be expressed as a weakness from the literature, but as a strength of this research to support other research and development. This paper is strong enough to avoid falling on the classic message about the need to fill obvious gaps in the literature.

AR1: We agree with the reviewer's point. Here we are referring to Beltaos and Prowse (2009) and it may be unclear. We have revised to make link to these authors more obvious.

Action1:  Page 3, Line 111: revise to say: "Specifically, these authors noted that broad scale…"

RC1: Line 168: Can you please double-check that Groudin is not Grondin (a more common name)? Also in Table 1. You may very well be correct.

AR1: confirmed it is Groudin

No Action1

RC1: Line 194: Is "potential" the right word here? My understanding of potential is what can be reached or achieved at a site or station, as opposed to the fine-scale maximum at a station for any given year.

AR1: agreed and removed word "potential"

Action1: Page 10, line 214-216: Revised to say: "These instantaneous values correspond with the water level at the initiation and maximum flood level for ice specific and open water conditions  during each calendar year."

RC1: Line 197: Should "daily" be "daily-averaged"?

AR1: WSC site reports values a "daily" time step. We opt to use: mean daily

Action1: Page 11, Line 220: Revise to say: "mean daily water level or mean daily discharge"

RC1: Line 199: Should "depends" be "depending"?

AR1: agreed

Action1: Page 11, Line 222: Revise to say "depending"

RC1: Figure 3: You could clarify this figure by adding the duration of the ice season. I am not sure that the title of the X axis is accurate. This cannot be a complete year, at least not if the scale is constant. Last B date is quite low compared with HM. Is this a typical behavior? I like that HO is significantly lower that HM, but again, is this typical? It just seems that so much water has been flowing during breakup and that the freshet is almost over by then. I understand that this may be representative of a specific river, but is this largely applicable / representative of Canadian River?

AR1:
-This 'conceptual schematic' was created by using Sept 1 to Aug 31 water level hydrograph from Mackenzie River at Norman Wells with mid-winter event superimposed over top. With the schematic, we are only trying to show spikes and rising water levels we look for when extracting data  with some vertical exaggeration and we did not make any mention of relative differences in magnitude between events.  We have revised caption to further clarify the conceptual diagram

Action1: Page 12, Line 247-254: Figure 3 caption revised to say:
"Figure 3. Conceptual schematic of continuous river water level hydrograph (black line) spanning September 1 to August 31. Period of ice affected flow is constrained by First B Date to Last B Date. A possible mid-winter break up event is shown as grey line, at approximate centre of hydrograph. Symbols for the 15 variables which populate the Canadian River Ice Database are shown in the figure (see Table 2 for additional information). The variables shaded in grey show the instantaneous water level and associated time when the event occurred  Compression of x-axis and vertical exaggeration of y-axis accentuates the water level changes observed during ice conditions.  The relative magnitudes of variables and water level pathology should not be considered as typical."

RC1: Figure 3: The peak to and from HM is intriguing to me. It is a relatively gradual rise, which does not suggest the formation of an ice jam. Then, the water level drop does suggest the gradual thermal melting of an ice jam. Also, in my mind, Last B date should be at higher level than HB, but I may be wrong.

AR1:  Figure 3 is simply a schematic presentation of ice affected river water level and largely aims to visually define the various parameters that are extracted for CRID; its appearance can change from site to site and from year to year as can the relative magnitudes of variables. To address this concern have added statement to caption

Action1: Page 12, Line 253: added statement:  "The relative magnitudes of variables and water level pathology should not be considered as typical"

RC1: Not sure if this is well positioned in Fig. 3. It seems that after freeze-up, thermal thickening or thermal erosion should follow. Therefore, I do not see why this first minimum Q would occur during the subsequent rise in water level. I may be wrong and you may have seen this at some stations.

AR1: What we are trying to show here is the possibility of different dates of minimum daily water level and minimum daily discharge. This is because open water stage discharge relationship is invalid during ice conditions.

Action1: Have moved HLQ1 closet to HLW1.

Action1: Page 12, Line 253: added statement: "The relative magnitudes of variables and water level pathology should not be considered as typical"

RC1: Table 2: First B date (and last B date): Has this been re-analyzed or indicated B dates were just adopted as they appeared in reports? My understanding is that B dates are often off by a few days and this can be checked with some temperature and hydrological indicators. It can have a significant impact when preparing flood maps that distinguish different flooding processes.

AR1: First B date and Last B date – input to CRID as they appear in the published NHP data and this is detailed in methodology. We gave full description of B date, applicable hydrometric manual references and some caveats.

No Action1

RC1: HF: This is quite obvious when the ice cover forms by frontal progression, but the gradual formation of border ice followed by ice congestion in a relatively narrow open water channel may not generate a clear signal. That being said, there would most probably always be a "maximum freeze-up level", and this may be a more appropriate name for this parameter. (I am unsure how you would differentiate that from a small runoff event taking place during freeze-up and generating or not, a freeze-up jam.). I appreciate the explanation provided at lines 362-368.

AR1: It is stated in paper that on occasion water levels crept up through the winter period as a result 'maximum' was removed from this variable name.

No Action1

RC1: HF2: Could change the name of this variable to "water level at second freeze-up"

AR1: A second freeze-up is only exclusive to a mid-winter break-up event. In case of water level creeping through the winter we observed maximums when assumed no break-up and refreezing of the river ice cover, so opted for a name that does not imply a process.

No Action1

RC1: Line 258: Drifting ice is part of the flow, it is not stagnant ice, and it should not generate bacwater if the surface concentration remains low. Same comment for flowing ice chunks.

AR1: This information is verbatim from Poyser et al, 1999. We are making a point that using B date alone does not tell much about specific river ice condition.

No Action1

RC1:Line 284: I am unsure why point C is not at the first spike that seem to be sharp enough to represent a local ice movement, possibly a downstream partial breakup that would reduce backwater at the station.

AR1: This figure is schematic published in Beltaos 2012. Point C was selected for illustrative purposes. That another spike was not selected is a good example of the 'judgement/art/subjective' aspect or extracting river ice information.  Notably, Beltaos has mentioned numerous time that Point C (synonymous to HB) is not the best metric.

No Action1

RC1: Lines 290-291: I believe that hydrological simulation, comparison with other stations, or judgment can still provide some kind or error margin (it can hardly be more than one order of magnitude, at least).

AR1: Thank you for paying attention to this. Suggestions on how to better estimate flow during this time are outside scope of this paper. To reduce confusion "error margin" is removed from text.

Action1: Page 17, Line 322-323: Revised sentence to say:  "Consequently, it is not possible to assign reliable flow estimates during this period, leading to the aforementioned "poor" characterization since there is no way at this time to quantify the reliability of these data."

RC1: Lines 307-311: Not sure if this paragraph invites CRID users to report on possible errors that could justify specific re-analyses. I believe that it should be the case, but it depends on how ECCC will want to maintain and update the CRID.

AR1:  We have revised the paragraph for more clarity with respect to data maintenance and updates. This aspect of CRID was also brought up by other reviewers. We address database errors and corrections at a later section.

Action1: Page 19, Line 349-351: Remove text  : "As a corollary, the water level interpretation toward the CRID research data set also required a high level of expert judgement with this subjective attribute inherent to the reported variables"

Action1: Page 19, Line 340 – 343: Revise to say "National assessments that analyze flow data often make no mention of the uncertainties associated with the collection and interpretation of hydrometric data during ice conditions (e.g. Cunderlink and Ouarda; 2009; Burn and Whitfield, 2016). More discussion on this issues are needed to inform the water community of the challenges related to cold-regions hydrometric data collection (Hamilton, 2003) and caution when interpreting study results"

Action1: Page 38, Line 720-723 : add sentence: "As is indicated on the Open Data Portal where the CRID can be downloaded, ongoing work with the CRID may include error checking and corrections, so users should use the latest version of the CRID by referring to the  version number that appears in the .csv file name (http://data.ec.gc.ca/data/water/scientificknowledge/canadian-river-ice-database/CRID_BDCGF_Versioning_EN_FR.txt).

Action1: Page 44 Line 857-861: add section: ". It is recommended that periodic updates be made to this database since a longer time series record is of more value. Based on the 160

locations in operation up to Dec 31, 2015 (Table A1), a 5 year update of CRID time series (2016-2020) would require 800 person-hours of work. Evaluation of future research priorities are needed to formalize whether this task would be completed by the same group or undertaken by others."

RC1: Line 327: The authors could confirm if this first B date on Oct 10 was the result of a rise in water level that trigger the decision to initiate B condition. More generally, the authors could confirm the information (cameras?) or signs (rise in water level after X degree-days of freezing) that are usually considered to initiate B conditions.

AR1: B date is decided by NHP and is an indication of ice affected flow. The example is used to illustrate that a channel wide, bank to bank ice cover is not present at gauge on the given B date. Our goal is not to validate First B, rather provide other less 'readily available' metrics of river ice which have some physical/process rationale. Page 18, Line 372 we did state "NHP reports"

No Action1

RC1: Figure 5: Adding the water level signal to this figure would be of interest, but it does represent some work.

AR1: Agreed and water level plot is added. This comment has also prompted a revision of the similar Figure 12 to include water level signal

Action1: Page 21-22: Figure has been revised to include water level signal. In addition remote sensing images perspective has been changed to overhead rather than oblique to aid visualization. Location of station now indicated with red circle. Figure caption has been revised to: "Figure 5.  Daily mean water level hydrograph for October 1 to November 15, 2000 at National Hydrometric Program gauging station Mackenzie River at Norman Wells (10KA001) along with MODIS time-lapse satellite images (accessed at: https://worldview.earthdata.nasa.gov/). Date of images corresponds with black arrow.  Station location indicated by red circle.  Width of the channel is approximately 1,300 meters and includes numerous islands. Flow is from bottom to top. First B Date is October 10 while freeze-over water level (HF) occurred November 9 and these images were obscured by clouds.   River channel open water is green and ice cover is white on these true colour images.."

RC1: Lines 351-352: The authors could mention something about peak factors (instantaneous divided by daily-averaged) here. For freeze-up, peak factors can be in the order of 1.1 or 2.0, depending on freeze-up dynamics... This would just be a reminder that using a 1 for design may be unsafe.

AR1: This is a good point and peak factors could be calculated from the CRID as it records both instantaneous and daily values at freeze-up whenever available.  Statement has been added with reference to peak factors at lines 365-368

Action1: Page 22, Line 428-429: revised to say: and (3) allow for calculation of peak factors (as a ratio between instantaneous and mean daily as described Zhang et al., 2005) to aid in design of river structures.

Have added the following reference:

Zhang, X., Buchberger, S.G., van Zyl, J.E. A Theoretical Explanation for Peaking Factors. World Water and Environmental Resources Congress. ASCE Library. https://ascelibrary.org/doi/10.1061/40792%28173%2951, 2005

RC1: Line 367: The authors could mention that this can take place over a distance of several hundred km upstream.

AR1: Agreed

Action1: Page 22, Line 427: Revise to say: "This process can take place over a distance of several hundred km upstream   (e.g."

RC1: Line 371: "...snow and..." First time I see this expression. In some regions, there is snow, but no ice cover because there is too much heat (downstream of lakes or reservoirs, or maybe downstream of cities and industries). Still, the word "snow" here may create confusion since this paper is about the ice cover period.

AR1: removed "snow"

Action1: Page 23, Line 433 – revise to say: "during the winter ice cover"

RC1: Lines 377-379: It could be mentioned that this is common and mostly caused by the thicker ice cover at the end of winter that generates a higher water level despite this being the actual winter min Q.

AR1: Agreed

Action1: Page 23, Line 440-441: Add statement:" This example illustrates how a thick, late winter ice cover would raise water levels due to reductions in channel cross sectional area."

RC1: Line 383: "analysis" is probable "analyses" (plural)

AR1: OK

Action1: Page 23, Line 445: change to "analyses"

RC1: Line 396: "risk" and "threats": The risk cannot be a threat. Consider rephrasing this considering that the risk is a combination of consequence and probability (or possibility) of a hazard and that a threat is in this case a hazard.

AR1: Thank you for the clarification. Have removed word "risk".

Action1: Page 24, line 458 Revise to say: "elevated water levels, and in extreme cases"

RC1: Lines 400-401: There are also records from nearby hydrometric stations.

AR1: This study did not evaluate nearby hydrometric station records.  This watershed continuum or watershed analog method would be a good way to verify if identification of perceived mid winter events was correct (Beltaos 1990, interpretation of these 'winter peaks' is a challenge).  CRID sites were treated independent for data extraction.

Action1: Page 25, Line 483-485: Changed to say: "Due to these inherent challenges of interpreting mid-winter break-up events, a closer examination of the CRID time series and comparison to nearby hydrometric stations may be required before pursing further analysis. "

RC1: Line 407: Not sure if a sudden drop in water level can be considered as a "spike". Also, depending on where the station is located and about the intensity of the winter runoff events, the water level signal can be a drop (local breakup) , a gradual rise (ice cover is lifted), a sudden rise (ice jam formation), or a combination of the above.

AR1:  The spike occurs as water levels increase above threshold for ice to become detached from banks and entrained in flow, resulting in reduction in hydraulic resistance. This is the characteristic "spike". We found the spike method to consistently appear on the rising water level limb at many sites. The drops in water level related to very thermal events which are not overly common in the CRID.

No Action1

RC1: Line 411: In areas where multiple mid-winter breakup events occur, they can be hard to distinguish from freeze-up chaos. First question: Does a mid-winter runoff event only qualifies after a complete ice cover has formed? Second question: why not using  the highest mid-winter peak instead of the first one? Third question: How would you consider a massive breakup event at the end of February like it happened in 1981 in southern and central Quebec? Would that be a mid-winter breakup followed by no more winter, or would that be the spring breakup event? I am curious, but understand that we may not have to start a conversation about this.

AR1: First Question Response: mid-winter runoff is assumed to have occurred after formation of ice cover. Second Question Response: We use highest mid winter peak (HMWM) but first initiation of mid winter breakup (HMWB). Third Question Response. We came across this issue in earlier work of Newton et al 2017 comparing Doyle 1984 mid-winter break-up events to earlier iteration of CRID. An event that we categorized as spring break-up event, Doyle categorized as a mid winter break-up event. This type of categorizing is a challenge since river ice is continuum.

No Action1

RC1: Lines 414-415: I am not too familiar with WSC's practice, but I would be very careful to remove a B in the middle of winter following a mid-winter breakup event. This may occur in NE, NB, southern QC and ON as well as in West-southern BC, but in most of Canada, after a complete mid-winter breakup, the presence of shear walls would prevent the removal of the B until the flow has receded significantly and this is when a cold spell may have already created border ice.

AR1: We agree that the text may be misleading so:

Action1: Page 25, Line 482: revise to say:  "extracting the mid-winter variables"

RC1: Line 431: "mark": I would say "may mark" as this is not the case for all types of rivers.

AR1: agreed

Action1: Page 26, Line 499: Revise to say "may mark"

RC1: Line 435: Depends where: In some cases, a mid-winter breakup event is followed by a dramatically cold period during which frazil generation is significant. The result may be a very thick ice accumulations, with inflated ice jams and new anchor ice cycles.

AR1: Thank you description of process. This is more appropriate to observation of multiple HMWB so added to previous section

Action1: Page 25, Line 475-477 :add "In some cases, a mid-winter breakup event is followed by a dramatically cold period during which frazil generation is significant. The result may be a very thick ice accumulations, more ice jamming and new anchor ice cycles".

RC1: Line 436: Of course, daily-averaged levels may appear smooth enough. At specific locations, the water level could remain high or even increase even though the discharge drops. This would be caused by progressive frazil accumulation produced in a newly open (steep) reach exposed to cold air. Hydrometric stations are usually not located in reaches affected by this type of process. I am just providing this information in case it would seem appropriate to adapt the text (and this applies to many other comments).

AR1: Thank you for description of process. We modify text as follows

Action1: Page 26, Line 504: revise to say "…generally reveal"
Action1: Page 26, Line 505-507: add sentence: "Notably, this patterns is likely typical on relatively flat river channels, while on steep river sections, progressive frazil accumulation produced in newly open section exposed to cold could increase water levels even during receding flows"

RC1: Lines 450-452: A hanging dam can form several km downstream of an open reach. It all depends on the river gradient and profile. In the case of anchor ice, it can hardly remain in place for several months. It will either contribute to the formation of a complete surface ice cover, or will melt away during mild spells and come back during cold spells. I suggest that this creeping signal is mostly associated with frazil acaccumulation.

AR1: hanging dams are very stable features and can remain in place for many months. We modify text as follows

Action1: Page 27, line 522: Add sentence: "However, anchor ice formations are not known to remain in place for several months"

RC1: Lines 452-453: But wouldn't it still deplete during the winter time? I see that you have a reference at the end of the sentence but does that reference suggest that?

AR1: We have no definitive information on how rapidly the depletion is for swamps and muskegs at this site.

Action1: Page 27, Line 524-525:  "though this assumes no depletion over the period of ice cover."

RC1: Figures 8 and 9: A superposed air temperature graph would be of interest.

AR1: We opted to not include air temperature data plots since reader may imply that a detailed evaluation of temperatures was part of this study. Temperature use was limited primarily to aid interpreting freeze-up (temps less than -10°C) and mid winter events (positive temps and rain).

No Action1

RC1: Line 471: Please update Figure #

AR1: Thank-you

Action1: Page 30, line 547: change to "Fig. 8"

RC1: Line 485: "Impure ice": Is this common? Should you explain what this means in brackets? Should you also add this to the previous sentence that refers to snow load, for consistency?

AR1: remove word 'impure' and 'snow load' from text and revise to say:

Action1: Page 31, Line 559-561: Revise to say: "since the specific gravity of river ice is commonly taken as 0.92. Nevertheless, these measurements are assumed to represent the actual ice cover thickness"

RC1: Lines 504-506: You may suggest that readers could take the measured thickness and associated date, evaluate the corresponding cumulated degree-days of freezing (or a cumulated sophisticated heat budget), and create a relationship between both parameters. Step 2 would simply be to apply this relationship to the maximum degree-days of freezing of each winter to obtain an estimate of the maximum ice thickness (if no midwinter breakup occurred between ice thickness measurement and actual max freezing degree-days).

AR1: Specific method of ice growth prediction is not within scope of paper so leave text as is.

No Action1

RC1: Lines 517-518: Actually, the station may start "feeling" some stage instabilities that come from upstream (these would actually be discharge instabilities induced by upstream ice movement), and it would still mean that breakup has initiated. How do we know that this is taking place downstream, especially when looking at daily-average stage data?

AR1: If only daily data, cannot determine HB. For clarity:

Action1: Page 34, Line 600: replace "in the absence of a continuous water level record." with "from a record of mean daily water level"

RC1: Line 518: Same comment as before: a reduction in roughness would generate a sudden drop of the instantaneous stage signal. In turn, a jave would be a spike and a sudden raise would be the formation of an ice jam downstream.

AR1:  Agree that a drop in water level, however, the method of pen chart reading assumes that water level rises, the ice cover detaches/ entrains, and then drops. Jave is mentioned at line 535.

No Action1

RC1: Line 529: I am not sure that there is a need to state "quickly" here. First, it applies to both time and distance traveled. Second, quickly is relative and I have seen large ice slabs (especially those that were part of a hanging dam or a snowmobile crossing) remaining fairly large several days or km after breakup.

AR1: Agree

Action1: Page 34, line 610: remove work "quickly"

RC1: Line 532: Should there be an example of a case study reporting X meters above the rating curve? This would illustrate the meaning of "far exceed"

AR1: Agree

Action1: Page 34, Line 614 -617. Revise to say: "open-water flow conditions (von de Wall et al, 2009, 2010; von de Wall, 2011) For example at Liard River near the Mouth (10ED001) the 25 year return period for ice affected water level was 16.11 m versus 9.69 m  for the open water event (de Rham et al, 2008a) "

RC1: Lines 534-535: This is not exact: They can also cause a measurable stage (actual discharge) depressions for several hours before reaching an equilibrium. The jave is much more sharp, especially in steep channels and when the released jam was not too far upstream from the station.

AR1: Agree

Action1: Page 34, Line 618-622. Revise to say: ". A jam lodged upstream of a guage can also have a measurable stage (actual discharge) depressions for several hours before reaching an equilibrium. The release of a jam can generate a sharp wave called a 'jave' (Beltaos, 2013) yet another dynamic mechanism that can generate the identified HM water level on instantaneous water level recordings)."

RC1: Lines 535: It should be stated that 1. Javes can only be adequately documented using instantaneous data. 2. Javes have probably been removed from discharge records  (at least in Quebec) as they were considered to be ice jams that had nothing to do with a discharge signal. It is also possible that javes and ice jams have been removed from some records because they were peceived as instrument pathologies. If there is enough evidence of this practice in some offices, the authors should mention it in the discussion.

AR1: agreed with point 1. Point 2. and 3. are suited to  future work that exclusively examines CRID time series for jave. Author recollection is that extreme spikes on water level recording charts were generally not filtered out by NHP and thus reported as instantaneous events.

Action1: Page 34, line 622: add statement "on instantaneous water level recordings"

RC1: Line 541: Could be completed by "... where the stage gradually returns to the stage discharge relationship as the discharge slowly increases"

AR1: agreed. The overall sentence has been revised

Action1: Page 35, Line 625-628: change to: "The less common "overmature" break-up sequence was observed at some CRID stations with no less obvious "spiking" of water levels. An example water level of with this occurrence characteristic on the Peace River in 1982 (Fonstad, 1982) is included in Beltaos (1990) where minor water level perturbations are followed by a generally smooth reduction to open channel conditions. . In some cases the HB and HM were interpreted to occur at the same time."

RC1: Line 550: Should the authors state that the last B date could likely be off by a few days? It is not to criticize the work done by different offices, but to warn users about this possible limitation. The last B date is specially difficult to confirm during thermal breakup years or when post-break ice runs from far upstream still occur after a complete local wash.

AR1: We are not attempting to quality control Last B date, rather inform the CRID includes alternative variables to the readily available last B date.

No Action1

RC1: Figure 12: Second image is very dark. Is there a way to tweak this?

AR1: Thanks you for the prompt.

Action1: Page 36, Figure 12: tweaked brightness and contrast of 2$^{nd}$ and 3$^{rd}$ image. Also took Reviewer recommendation on Figure 5 to add water level record. Thus, the Figure caption has been revised to: "Figure 12: Continuous 15 minute interval water level hydrography for April 15 to 30, 2010 at National Hydrometric Program gauging station Hay River near Hay River (07OB001) along with images courtesy of Alberta Research Group. Left: Image looking upstream taken 7 days prior to spring break-up initiation (HB) of April 24, 2010, 04:25. Channel width of approximately 63 meters. Centre, left is a night time image 5 minutes after HB and shows evidence of fragmented ice in the channel. Centre, right is 65 minutes after HB and shows channel nearly clear of ice. Right image is 5 minutes after maximum spring break-up water level on April 25, 2010, 15:25. Stranded ice on channel banks indicates higher water levels. Last B date was April 28, 2010."

RC1: Line 575: Should the authors mention that Ho may actually occur in mid-summer (e.g., Saguenay event in Quebec, 1996) or during the fall, and therefore may not be associated with the spring freshet, especially in Eastern Canada?

AR1: Agreed

Action1: Page 35, Line 670-671: Add sentence "A Canadian perspective on flood process (snowmelt, rain-on-snow, rainfall) and their seasonality are detailed in Buttle et al., (2016)."

RC1: Line 577-578: Just to complete the idea, i would suggest: "...for a large ratio of hydrometric stations in Canada, and most probably for an equal ratio of unmonitored sites."

AR1: Agreed though use word "portion" instead of "ratio"

Action1: Page 37, Line 670: Add "for near one third of hydrometric stations in Canada (e.g. von de Wall 2009) and most probably for a similar proportion of unmonitored sites"

RC1: Line 584: "five" should probably be "six"

AR1: thank-you

Action1: Page 37, Line 678. Change to "six"

RC1: Line 605: Should the author specific what defines an error or what it the calculation behind this %?

AR1: OK. This section has comments from all reviews so revising text for clarity.

Action1: Page 38, Line 699-708: " A quantification of human error in transcribing CRID data was undertaken using automated scripts to extract and compare the CRID daily discharge and First and Last B Date to those published by the NHP. Daily discharge was incorrectly transcribed on 4.7% to 7.8% of the time series depending on the variable while mid-winter associated discharge had the highest input error at 16%. This higher percentage of error is a likely remnant to the multiple rounds of revisions to mid-winter time series and confusion that arises when examining non-consecutive events that can occur across calendar years. For ice seasons when both a First and Last B Date were available, dates were incorrectly transcribed on 7.5% of time series.  All erroneous daily discharge and First and Last B Date values were replaced.  The remaining CRID data entries are not amendable to automated quality control since they were manually extracted."

RC1: Lines 606-607: It is unclear to me if indicated B dates are considered true and other parameters are corrected consequently, or the opposite.

AR1: OK. Text revised for clarity:

Action1:  Page 38; Line 705-706. Revise to say: "For ice seasons when both a First and Last B Date were available, an input error ofdates were incorrectly transcribed on  7.5% was found.of time series

RC1: Lines 613-614: As asked earlier, would the authors also commit to present updated versions of the CRID with corrections?

AR1: Thank you for reiterating

Action1: Page 38, Line 720-723: added statement "As is indicated on the Open Data Portal where the CRID can be downloaded, ongoing work with the CRID may include error checking and corrections, so users should use the latest version of the CRID by referring to the  version number that appears in the .csv file name (http://data.ec.gc.ca/data/water/scientificknowledge/canadian-river-ice-database/CRID_BDCGF_Versioning_EN_FR.txt).

RC1: Lines 623 vs. Line 634: If I had to choose, I would say that ice processes are site specific.

AR1: Agreed

Action1: Page 39, Line 733 change to "are"

RC1: Line 628-629: (e.g. promoting a thicker ice cover in the deck shadow and promoting ice jamming against abutment or pillars)

AR1: Thank you for this addition. Use word piers instead of pillars.

Action1: Page 39, Line 739: Add to end of sentence: "such as promoting a thicker ice cover in the deck shadow and promoting ice jamming against abutment or piers"

RC1: Figure 13: The legend in this graph could include variable acronyms for clarity. Also, it would have been useful to separate the two populations with different icons / colors. The only obvious difference is the two populations is blue circles.

AR1: Agreed. Revise figure and associated text as follows:

Action1: Page 40, Figure 13: Figure has been modified following suggestion, the caption has been revised

Action1: Page 39, Line 745: added: "towards assessments of station homogeneity are a necessary next step"

Action1: Page 39, Line 747: added: "this rudimentary visualization of data towards confirming non-homogeneity reveals the"

RC1: Table 4: There may not be enough space, but the authors could consider adding a column with the variable acronym.

AR1: Agreed and there is enough space

Action1: Page 42, added added column 'Symbol' to table

RC1: Line 694: "Very often" Do we have an updated number about that? If not, I hope that the CRID will be used by researchers to update the one third presented by Beltaos years ago.

AR1: von de Wall (2009) is most recent and has been added to this statement.

Action1: Page 43, Line 814-815: Revise to say "It has been established that extreme flooding in ~ 30% of Canadian rivers is often the result of ice processes and jamming (Beltaos, 1984; von de Wall, 2009)"

RC1: Line 694-695: I am not sure that I agree with this interpretation. It can be said that ice jams produce higher water levels at similar high flows (quite logical), and it can be said that at some sites, the main flooding process is caused by ice processes. In turn, the highest discharge in rivers most often occur in the absence of ice. There should be a more efficient way to express this.

AR1: As written it reference to Gerard 1989 and statement is revised to address reviewer concern:

Action1: Page 43, Line 816-817: Revise to say "At these locations stream discharge cannot be used to quantify flood level since the stage-discharge relationship is invalid during ice conditions"

RC1: Line 696: "eg." should be "e.g.". I take note that FloodNET is only one example. Other groups have completely ignored river ice processes in their flood research.

AR1: Agree with change. Don't have specific ref for Ouranos and Global Water Futures, but the groups are mentioned in Turcotte et al, 2019 will revise

Action1: Page 43, Line 820-821, revise to say: "(e.g. NSERC FloodNet, 2015, other groups mentioned by Turcotte et al., 2019), likely as a result of the limited, long term field"

RC1: Line 700: "could likely" should be "should, when applicable,"

AR1: agreed

Action1: Page 43, Line 825: Revise to "should, when applicable,"

RC1: Line 701: For sites that are not included in the CRID and where winter water level information is available, the CRID can represent a template to extract pertinent information for various purposes, including flood mapping and hydraulic structure design.

AR1: Thank-you for this. Have revised text.

Action1: Page 44, Line 856-857. Add sentence "For sites not included, the CRID can represent a template to extract pertinent information for various purposes including flood mapping and hydraulic structure design"

RC1: Line 742: A last sentence could be: "Maintaining funding and constantly improving hydrological estimation and measurements approaches is needed to maintain an adequate level of knowledge and to update the CRID in the future."

AR1: Since CRID was completed using public service tax dollars not appropriate to make call for additional funding. Hydrological estimation and measurement approaches are outside the scope of this data description paper of the CRID. From earlier reviewer comments a statement was added about updates (page 44, line 858 to 861): "Based on the 160 locations in operation up to Dec 31, 2015 (Table A1), a 5 year update of CRID time series (2016-2020) would require 800 person-hours of work. Evaluation of future research priorities are needed to formalize whether this task would be completed by the same group or undertaken by others". We do agree that associating last sentence to CRID and river ice science is a good idea so have added :

Action1: Page 45, Line 872-873: The CRID supports continued research on river ice processes and the extreme water level fluctuations common to many cold regions river systems.

RC1: Lines 1042-1044: I do not see this paper referred to in the paper and it should removed from the reference.

AR1: Apologies on the oversight. This paper (Turcotte et al., 2019) is now referenced in text

Action1: Referenced page 42, line 839 "…other groups mentioned by Turcotte et al, 2019),…"

Title: A Canadian River Ice Database from National Hydrometric Program Archives

Author(s): Laurent de Rham, Yonas Dibike, Spyros Beltaos, Daniel Peters, Barrie Bonsal, Terry Prowse

MS No.: essd-2020-29

RC2 General Comments:

The manuscript introduces the newly developed Canadian River Ice Database (CRID). Such a database is very welcomed in the river ice science and practitioner community and will promote studies to address a variety of research questions and practical issues. It is tremendous efforts to go through the large amount of historical data and collect the variables related to specific key ice events. Several of these variables can be very challenging to identify and require extensive expertise in river ice engineering, which is offered by the author team. The team's experience and expertise are also reflected in the selection of the variables, detailed description of their physical importance, quality control of the ice data, and uncertainty assessment. In this regard, the manuscript provides an important reference document for the use of the CRID. I will definitely be using the database and would like to see it being updated regularly as new information becomes available.

AR2: Thank you for this overview and positive feedback on the work. The note about 'uncertainty' has initiated authors to undertake the following:

Action2:  Page 9, Line 191-193. Added sentence: "A final note: the vast majority of historical annual water levels (item (8)) are reported by NHP as preliminary since these values were never published. Similarly, some recent digital water level files (item (2)) were also preliminary since NHP had not yet screened these data."

Action2: Page 37, Line 675: have added "Uncertainty" to the section title.

Action2:  Page 37, line 687-690: added sentence: "The vast majority of mean daily water level pages and some of the more recent digital water level recordings were deemed "Preliminary" by NHP. Different methods of collecting requisite information for mean daily water level have existed over the archive from at site station observers who viewed a staff gauge once daily to the more modern arithmetic averages determined from continuous water levels."

Specific comments:

RC2 Line 87: select to selected

AR2: Agreed

Action1: Page 3, Line 100. Change to "selected"

RC2 Line 115-126: It seems that with minimum 20-year record, no minimum drainage area and including both north of 0deg isotherm and southern temperate zone would result in much more than 196 stations. Am I missing any additional selection criteria used here?

AR2: line 122 authors state 'subset' and Line 128/129 'sites prone to mid-winter break-up events'. The text references (Prowse and Lacroix, 2001) from which the initital subset was selected.

Action2: Page 4, Line 129: have added "the near 8,400 active and discontinued"
Action2: Page 4, Line 132. Changed sentence to being with: "These select"
Action2, Page 4:, Line 139: add statement: "Inclusion of these sites resulted in a network of"

RC2 Line 135: foci to forcing

AR2. Foci is s common term in ecology, though other reviews mentioned confusion with "foci". For clarification:

Action: Page 3, line 150: change "foci" to "focus"

RC2 Line 139: listing to list

AR2: agreed

Action2: Page 5, line 154 change to "list"

RC2 Line 191: There are actually more than 15 variables as several of the ones listed in Table 2 include both water level and discharge and they probably should be counted as 2 variables.

AR2: We use 'variable' in a multidimensional sense to include all data types associated with each variable: water level, discharge, date, time, rating.

No Action2

RC2 Figure 3: I am not sure if this figure is based on actual gauge record or purely conceptual. It may worth to show a water level hydrograph where the key ice events are less obvious (less "spiky") and explain how the different variables are identified.

AR2: As the caption states, the figure is conceptual schematic.  It was based on actual water level record in the Mackenzie River at Arctic Red and we added a mid winter section. Questions about this figure were also brought up by the other two reviewers.  Given all of these comments, Figure 3 caption was revised.

Action2: Page 12, Figure 3, line 247-254.  Caption has been revised to address these concerns and was described in  Reviewer 1 comments

RC2 Table 2: does the wording "data accuracy" best represent what this indicator really means? It may lead reader/user to think the published data is accurate while it is less likely in case of ice affected discharge data.

AR2: Good comment. Data resolution is better representation

Action2: Page 11, Line 218: change accuracy to "resolution"
Page 11, Line 221: change accuracy to "resolution"
Page 15, Line 257: Table 2 caption: change Accuracy to "Resolution"
Page 17, Line 296:  add "(Table 2: "Discharge" under column "Data Resolution").  "

Page 34, Line 602: change accuracy to "resolution
Remainder of MS has been checked and confirm no other changes required

RC2 Section 3.3 What are the methods used to compute discharges under ice conditions? Can the authors briefly describe some common ones? This is important information for users of the published discharge data. Additionally, my understanding is that different methods and techniques have been used when deciding when to start and end the B symbol. Maybe the authors can provide some information on this as well?

AR2: Section 3.3,  Page 16, Line 276-278 states:
"This section highlights challenges related to data collection during the ice season through excerpts from hydrometric program operational manuals, other publications and experience in developing this database. This background information is considered of high value to users when interpreting spatial and temporal characteristics of river ice."
We are only attempting to provide background information rather than explain the intricacies of under ice discharge estimates and B dates. We included references and appropriate statements so readers can inform themselves:
Line 289: Poyser et al (1999) is referenced and have listed the types of river ice conditions that can result in B date.

Here is reference:
Poyser, B., Leblanc, R., and  Kirk, D: Lesson Package No. 20 – Computation of Daily Discharge (Ice Conditions),  The Water Survey of Canada, Hydrometric Career Development Program, 1999.

Line 291:  verbatim statement of Environment Canada 1980 with several methods to compute discharge under ice. Reader can look at reference for specifics on how discharge is calculated.

Here is reference:
Environment Canada: Manual of Hydrometric Data Computation and Publication Procedures, Fifth Edition, Inland Waters Directorate, Internal Report, Ottawa, 1980.

No Action2

RC2 Line 278: repetitive quotation marks

AR2: OK

Action2: Page 17, Line 310: Revise to say: "

RC2 Page 12: Section goes from 3.3 to 3.4.1, missing 3.4

AR2: OK. Thank you for picking up this missed detail

Action2: Page 19, Line 357 add section: "3.4 CRID Variables"

RC2:Line 345-348: It may not be accurate to say the initial ice cover progression past a gauge is always a spike in the water level chart. In many cases, the "stage up" caused by an ice cover approaching from downstream and passing a gauge is a gradual water level increase. How is HF decided in a case like this?

AR2: Since we had access instantaneous water level recording we examine for rise and maximum  in water level to indicate possible start of bank to bank ice cover.  We do state at page 12, line 421 "Beltaos (1990) discussed the unlikelihood that a complete ice cover forms at the instant of HF." We acknowledge the use of work 'spike' is not a good describer so revise text as follows:

Action2: Page 22, Line 406. Revise to say "This initial ice cover progression upstream past the gauge can cause a gradual increase to a maximum in the water level chart and is depicted as HF (freeze-over water level) in Fig. 3."

Action2: Page 22, Line 418: delete "freeze-up spikes" and change to "maximum freeze-over water level"

RC2 Line 483: maybe add "approximately" before 0.92 as ice density can be affected by many factors.

AR2:  Review 1 also brought up this item and text was revised.

Action2: change text page 30, line 559 to "since the specific gravity of river ice is commonly taken as 0.92"

RC2 Line 501 Fig. 10 should be Fig. 11

AR2: thank-you

Action2: Page 33, Line 579 change to  "Fig. 11"

RC2 Line 517-518: this statement about the spike on the water level hydrograph indicating the onset of breakup seems to be conflicting with line 539-541. In the case of thermal breakup, how is HB determined?

AR2:  Agreed this needs clarification. The following revisions to text are detailed below:

Action2: Page 35, line 625-626 change no obvious to "less obvious"
Action2: Page 35, Line 627-628: added "where minor water level perturbations are followed by a generally smooth reduction to open channel conditions. In some cases the HB and HM were interpreted to occur at the same time."

RC2 Line 529-531: ice jams can form at morphologically conducive locations even without intact ice cover stopping the ice run.

AR2:  Unknown occurrence to authors so revise as

Action2: Page34, Line 612-613:  Added sentence "According to an anonymous reviewer, ice jams can also form at morphologically conducive locations even without an intact ice cover stopping the ice run"

RC2 Line 534-535: Jams formed upstream of a gauge may also choke the flow. It also depends on its vicinity to the gauge.

AR2: RC1 had similar comment

Action2: Page 34, line 618-622: added sentence: "A jam lodged upstream of a guage can also have a measurable stage (actual discharge) depressions for several hours before reaching an equilibrium. The release of a jam can generate a sharp wave called a 'jave' (Beltaos, 2013) another dynamic mechanism that can generate the identified HM water level on instantaneous water level recordings."

RC2 Line 545 chuck -> chunk

AR2: OK

Action2: Page 35, line 632: Revise to say: "chunk"

RC2 Line 553-556: I wouldn't say the last B date is always used as a surrogate/index, and less accurate than the CRID data to analyze spring breakup timing. They just represent different stage of the breakup.

AR2:  We said the last B date is **sometimes** used not **always** used. In any case, the Last B date is final day that ice affects channel flow condition at the gauge, however, there may be no actual ice at gauge, and rather, the flow condition is affected by  backwater from ice downstream. In general the sequence and processes associated with ice break-up all occur prior to the Last B date. However, this would depend on specific river flushing and clearance characteristics at the gauge. Users of data should view Poyser et al 1999 which is WSC publication describing discharge estimates under ice.

No Action2

RC2 Line 573-575: how can one calculate the water level using rating curve when instrumentation is damaged or not functioning?

AR2: Only discharge values are estimated, generally by interpolation and indicated with "E" by NHP to indicate that it is an estimate. Word calculate is misleading so:

Action2: Page 37, line 666: remove calculate and replace with "estimate"

RC2 Line 603-607 unclear to me how the percentage error are calculated.

AR2: Human input error versus NHP reported value as extracted by automated script. This section was also unclear to other reviewer so revised section.

Action2: Page 38, Line 699-708 as follows:

"A quantification of human error in transcribing CRID data was undertaken using automated scripts to extract and compare the CRID daily discharge and First and Last B Date to those published by the NHP. Daily discharge was incorrectly transcribed on 4.7% to 7.8% of the time series depending on the variable while mid-winter associated discharge had the highest input error at 16%. This higher percentage of error is a likely remnant to the multiple rounds of revisions to mid-winter time series and confusion that arises when examining non-consecutive events that can occur across calendar years. For ice seasons when both a First and Last B Date were available, dates were incorrectly transcribed on 7.5% of time series.  All erroneous daily discharge and First and Last B Date values were replaced.  The remaining CRID data entries are not amendable to automated quality control since they were manually extracted"

RC2 Line 635 Fig 12 should be Fig. 13

AR2: OK

Action2: Page 39, line 745: change to "13"
The authors developed a Canadian River Ice Database using the Canadian National Hydrometric Program hydrometric records. River ice related events, especially ice jam flooding, are of great importance to the watershed management in many cold regions around the world, including Canada. This database provides a significant amount of valuable data to support river ice research and applications. I can definitely see myself and my colleagues using this database. This paper is well organized and well written. I only have some minor concerns as indicated in the comments to the authors. I suggest a minor revision.

RC3: We thank Dr.Li for her comment and valuable review.  It is encouraging that she highlighted the importance of river ice for watershed management and the CRID data as presented is valuable for research and applications.

RC3: Detailed Comments to the Authors:

RC3:- Line 24: "73,000 variables" should be changed to "73,000 records".

AR3: This a useful comment. Rather than change to records, which will not be consistent with remainder of paper, for clarification we change to:

Action3: Page 1, line 25: change text to "73,000 recorded variables"

RC3:- Line 28: "a time series of up to 15 variables" should be changed to "time series of up to 15 variables".

AR3: agreed

Action3: Page 1, line 29: revise to say "time series of up to 15 variables"

RC3:- Lines 119-126: It is not clear how the 196 sites in this database were selected. Does it include any of the additional 60 southern sites? Or is it the same 196 gauging stations as in the NHP archives?

AR3: The paragraph describes the evolution of this subset of 196 NHP gauging locations and similar questions came up from RC2. Paragraph has been modified to quantify total number of active and discontinued NHP stations and highlight "These select monitoring sites".  Have added following statement for clarity:

Action3: Page 5, Line 139:  Have modified final sentence to  "Inclusion of these sites resulted in a network of  196 sites with drainage areas ranging from 20.4 km2 to 1.68 x 106 km2, including both natural and regulated flow conditions, with the latter distributed throughout this range."

RC3:- Line 135: Typo: "hydro-ecological foci".

AR3: This is terminology is used in hydo-ecological studies but since other reviewers also commented the text is now revised:

Action3: Page 5, Line 150. Change from "hydro-ecological foci" to "hydro-ecological focus"

RC3- Figure 2: Consider removing the border lines and using a different color for stations not in operation.

AR3: OK

Action3: Page 8, Figure 2:  border line has been removed and color when station are not in operation has been made darker

RC3- - Table 1: Add bottom border.

AR3: OK

Action3: Page 10, Table 1. Added bottom border

RC3:- Figure 3: Add a legend for the grey line to show it is the water level during mid-winter breakup.

AR3: Grey line is described in the caption so does not need to be shown as a legend item.

No Action3

RC3:- Line 265: It's not clear which 12 discharge time series the authors meant.

AR3: OK. In addressing this comment it was determined that this number should be 11. We also clarified in text by referring to location on Table 2.

Action3: Page 17 Line 295-296: revised to say: "for the 11 reported at-site ice affected discharge time series. (Table 2:  "Discharge" under column "Data Resolution")"

RC3:- Line 315: The subtitle of section 3.4 is missing.

AR3: OK. Addressed for RC2

Action3: Page 19, Line 357 added "3.4 CRID Variables"

RC3:- Line 325 & Figure 5: Consider defining the colors in the MODIS images for readers who are not familiar with satellite images.

AR3: OK. These images are true colour.

Action3: Page 22, Line 399-400 added sentence: "River channel open water is green and ice cover is white on these true colour images"

RC3:- Line 333: An extra space in "Sect. 3.4.4 )".

AR3: OK.

Action3: Page 20, Line 380 remove extra space

RC3:- Line 365: "parameterizes" should be changed to "parameterize".

AR3: OK.

Action3: Page 22, Line 424 change to "parameterize"

RC3:- Line 466: An extra space in "level ."

AR3: OK

Action3: Page 30, line 542. Change to "level."

RC3:- Line 496: No need to provide the abbreviation S.T.B. if it is used only once in the manuscript.

AR3: OK

Action3: Page 32, Line 575. Remove "S.T.B"

RC3:- Line 512: An extra space in "(84 days after January 1) ."

AR3: OK

Action3: Page 34, Line 592 remove extra space

RC3:- Line 618: An extra space in "about 1 hour ."

AR3: OK

Action3: Page 39, Line 728 remove space

RC3:- Table 2: Change "2000-01" to "2000-2001".

AR3: OK

Action3: Page 10, Table 1: Revise to say: "2000-2001"

RC3:- Tables 2 and 3: The column heads need to be re-formatted.

AR3: OK

Action3: Page 14 and 15: Remove line gaps on column head for Table 2 and 3. Also add Line at bottom of these tables.

RC3:- Lines 365-368: It is not quite clear why the length of water level data was determined to be 30 days.

AR3: OK

Action3: Page 22, Line 425-426. Revise to say: "tabulates water level for 1 month as"

RC3:- Line 412: What about HMWB? How was it determined when there are no continuous water level records?

AR3: Cannot determine HMWB in absence of instantaneous records. This is good observation and have removed "D" (Daily) from Water level and Time column in Table 2

Action3: Page 25, Line 472. Added sentence: "This variable cannot be determined from mean daily summaries of water level records."

RC3:- In Section 3.4, the variables were classified into 7 groups (7 subsections). Reasoning for the classification should be provided and reflected in the subtitles.

AR3: Thank-you.

Action3: Page 19, line 359-361. Moved sentence from above paragraph to below heading '3.4 CRID Variables: and state: "The following sub sections, corresponding to the four seasons of occurrence (Table 2) provide the background, extraction details and justifications for the selected CRID variables. For ease of reference the ice cover season is divided into three subsections that describe a maximum of four variables."

RC3:- A brief data management plan, particularly the current database maintenance and update plan, should be provided.

AR3: Thank-you. This is common theme from all reviews. It has been addressed as follows: conclusion:

Action3: Page 38, Line 720-723. Added sentence. "As is indicated on the Open Data Portal where the CRID can be downloaded, ongoing work with the CRID may include error checking and corrections, so users should use the latest version of the CRID by referring to the version number that appears in the .csv file name (http://data.ec.gc.ca/data/water/scientificknowledge/canadian-river-ice-database/CRID_BDCGF_Versioning_EN_FR.txt)."

Action3: Page 44, Line 857-861: "It is recommended that periodic updates be made to this database since a longer time series record is of more value. Based on the 160 locations in operation up to Dec 31, 2015 (Table A1), a 5 year update of CRID time series (2016-2020) would require 800 person-hours of work. Evaluation of future research priorities are needed to formalize whether this task would be completed by the same group or undertaken by others."

RC3:- There are some minor formatting errors in the references section. For example, the format of doi is not consistent. All references should be provided in the same format.

AR3: We have gone through the reference section and made formatting corrections to maintain consistency. Final formatting corrections will be made by the journal at the final editing stage.

Action3: went through references to ensure all doi format begins with http or https

[revised manuscript text omitted]